# Metasurface-empowered snapshot hyper-spectral imaging with convex/deep (CODE) small-data learning theory

Chia-Hsiang Lin [1,2,5] ✉, Shih-Hsiu Huang[3,5], Ting-Hsuan Lin [1] &
Pin Chieh Wu [3,4] ✉

Hyperspectral imaging is vital for material identification but traditional systems are bulky, hindering the development of compact systems. While previous metasurfaces address volume issues, the requirements of complicated fabrication processes and significant footprint still limit their applications. This work reports a compact snapshot hyperspectral imager by incorporating the meta-optics with a small-data convex/deep (CODE) deep learning theory. Our snapshot hyperspectral imager comprises only one single multi-wavelength metasurface chip working in the visible window (500-650 nm), significantly reducing the device area. To demonstrate the high performance of our hyperspectral imager, a 4-band multispectral imaging dataset is used as the input. Through the CODE-driven imaging system, it efficiently generates an 18-band hyperspectral data cube with high fidelity using only 18 training data points. We expect the elegant integration of multi-resonant metasurfaces with small-data learning theory will enable low-profile advanced instruments for fundamental science studies and real-world applications.

Optical imaging technology utilizes light to reproduce the form of objects via 2D images. In addition to spatial information, spectral imagers can provide more insights by introducing an additional dimension into 2D color images. Thus, they have become vital systems for science and real-world applications like metasurface-assisted miniaturized satellite remote sensing[1,2]. In general, the obtained multispectral and hyperspectral (categorized by the number of spectral channels) images are 3D data cubes, which map the irradiance of objects onto two spatial axes and one spectral axis[3]. Nowadays, hyperspectral imaging has been widely used in numerous areas, including agriculture[4], space communication and imaging[5], surveillance[6], and biotechnology[7].

One approach to constructing the 3D hyperspectral data cube is to sequentially acquire either spectral (with tunable filter elements)[8] or spatial (with dispersive elements)[9,10] scanning measurements that different scanning methods can accomplish. However, the scanning process is usually time-consuming, which is highly undesirable, particularly in scenes with relatively high-speed motions. In contrast, hyperspectral imagers with single shot mode (also called snapshot)[11] collect the comprehensive data within a single integration period, significantly reducing the time for data cube construction. Nevertheless, the requirement of bulky elements like mechanically tuning parts in scanning imagers and several dispersive components in conventional snapshot imagers makes both systems cumbersome, limiting the range of applications.

In the past decade, resonant metasurfaces have been proposed to address issues in conventional optical components[12,13]. Metasurfaces are regarded as arrays of subwavelength meta-atoms, which enable abrupt changes to the electromagnetic amplitude, phase, polarization, etc., of scattered light at the nano-scale. Therefore, metasurface opens

[1]Department of Electrical Engineering, National Cheng Kung University, Tainan 70101, Taiwan. [2]Miin Wu School of Computing, National Cheng Kung University, Tainan 70101, Taiwan. [3]Department of Photonics, National Cheng Kung University, Tainan 70101, Taiwan. [4]Center for Quantum Frontiers of Research & Technology (QFort), National Cheng Kung University, Tainan 70101, Taiwan. [5]These authors contributed equally: Chia-Hsiang Lin, Shih-Hsiu Huang. ✉e-mail: chiahsiang.steven.lin@gmail.com; pcwu@gs.ncku.edu.tw

up opportunities in increasing controllable degrees of freedom of light in a compact form. Metasurface-based imaging technologies, including hyperspectral imagers, have also been realized as a promising nanophotonic platform for low-profile optical systems. One of the pioneering works for light-field imaging utilized an achromatic all-dielectric metalens array for acquiring 3D information in the visible[14]. In that work, the spectral signal is absent after imaging post-processing, i.e., only the depth information is obtained. A metasurface-based hyperspectral imaging for biodetection has also been reported[15]. Although promising for spectral data collection, the signal computation requires the resonant wavelength shift so that the analyte has to attach to the nanostructures physically. By analogy with a push-broom imager, Faraon et al. demonstrated a compact hyperspectral imager with four dielectric metasurfaces[16]. The imager acquires the dataset by line scanning the object, requiring an extended measurement period. The need for a sophisticated alignment fabrication process in such folded metasurface design also increases the difficulty in device preparation[17]. To efficiently obtain the data cube, Arbabi et al. reported a snapshot spectral imager by incorporating the concept of multi-aperture filtered camera with meta-optics[18]. However, it is very challenging to approach a broad working bandwidth with a high spectral resolution by utilizing high-Q resonant filters. Recently, an ultra-spectral imaging device has been demonstrated for real-time biodetection[19]. Despite the high spectral resolution, an extensive metasurface structure library has to be preliminarily established for the dynamic imaging chip. In addition, using a compressive sensing algorithm can trade the spectral reconstruction error and spatial resolution off against the computational time determined by the number of microspectrometers.

In this work, we propose and experimentally demonstrate a low-profile snapshot hyperspectral imager by taking advantage of flat meta-optics and computational imaging with a small-data learning theory (see Fig. 1). The advanced metasurface-driven hyperspectral imager is realized using a specifically designed multi-wavelength off-axis focusing meta-mirror (MOFM) constructed by multi-resonant plasmonic meta-atoms. Working in the visible window, the MOFM

enables acquiring a multispectral dataset of 4 images in a one-shot measurement. We emphasize that the multi-imaging channels are obtained using a single metasurface chip, which was previously challenging and effectively addresses the issue of a large device footprint. The number of spectral channels will be extended to 18 via computational imaging using an innovative CODE small-data learning theory (only 4-band images are needed as the input) inspired by the convex optimization (CO) method in conjunction with deep learning (DE). Indeed, combining metasurface devices and computational imaging techniques has been proposed for full-color imaging applications[20–23], in which the images' quality is improved after post-processing. Unlike forward design approaches, DE provides an efficient platform to design nanophotonic structures[24] and retrieve spectral/spatial information[25], even for those missing in the image capture process. However, DE requires extensive data libraries (usually > 10, 000 data points are needed) for abstraction data learning[26]. Nevertheless, collection of big data like ground truth of hyperspectral images is no picnic, although the DE does not rely on advanced mathematics. On the contrary, CO works well even with small/single data at the cost of the math-heavy algorithm design procedure. Considering that extensive data collection and math-heavy derivation are daunting tasks for most software engineers, we proposed a machine learning/imaging theory by blending CO and DE. Our CODE theory was initially developed for hyperspectral satellite imaging image restoration, archiving advanced satellite missions using concise mathematical algorithms and small data. Even with inputs of aberration-caused blur images, the CODE theory perfectly transfers the 4-band multispectral imaging into an 18-band hyperspectral data cube with high fidelity, as illustrated in Fig. 1.

## Results

### Design of multi-resonant meta-atom for multi-wavelength off-axis focusing meta-mirror

The main idea for constructing a multispectral image of a color object without involving filters is utilizing a multi-resonant meta-atom as the metasurface building block. We incorporate an Al nano-rod with a

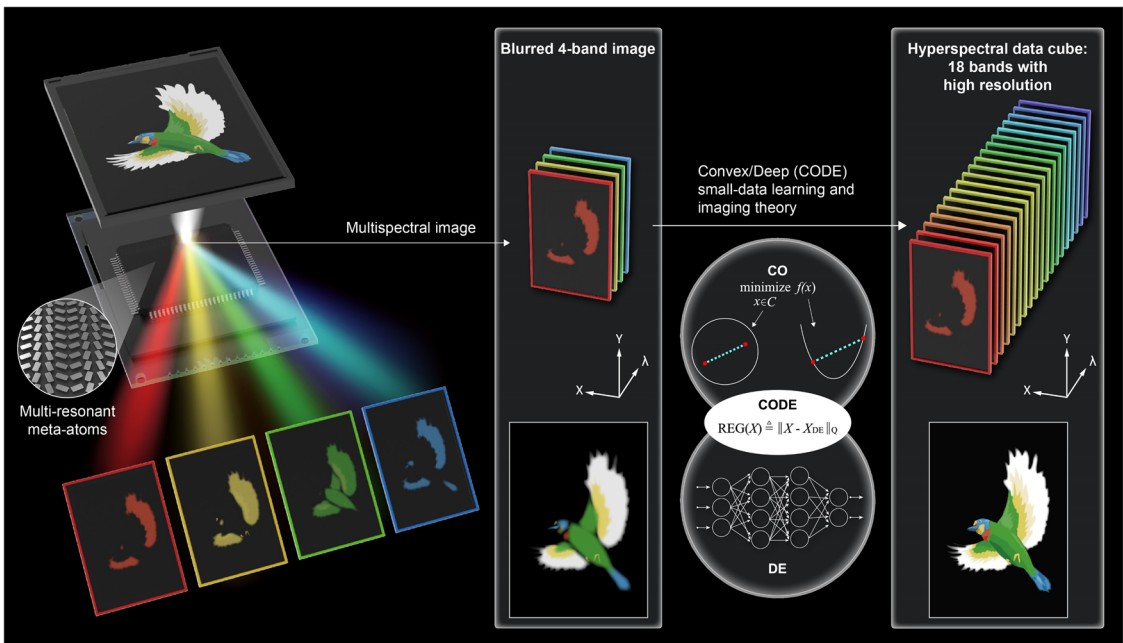

**Fig. 1 | Conceptual sketch of metasurface-empowered hyperspectral imaging.** The developed off-axis focusing meta-mirror, composed of multi-resonant meta-atoms, can simultaneously image a color object and spatially separate the image into four wavelength channels in free space. The acquired 4-band multispectral image will be used to construct an 18-band hyperspectral data cube using the CODE small-data learning and imaging theory inspired by the convex optimization (CO) in conjunction with the deep learning (DE) technique.

particularly designed distributed Bragg reflector (DBR) to realize a multi-resonant meta-atom. Details of the DBR substrate can be found in the Supplementary Table 1. The anisotropic shape of the Al nano-rod can satisfy the geometric phase condition[27,28], while the combination of plasmonic nanostructures and a DBR substrate might generate high-Q Tamm resonances[29]. To generate multi-resonant peaks across the spectral range of 500–650 nm, the thickness of constitutive layers in the DBR substrate is gradually varied, which is entirely different from previous studies in which the DBR is constructed by two alternating layers respectively with a fixed thickness[30]. Figure 2a shows the circular cross-polarized spectrum of the optimized Al meta-atom, presenting multiple reflection peaks >75% across the spectral range of interest. As can be seen in Fig. 2b, c, by rotating the topmost nanostructure, the phase shift of each peak wavelength can be continuously modulated from 0 to 2π with a bit of reflection intensity variation. While the circular cross-polarized reflection in a geometric phase metasurface ideally remains constant irrespective of the structural rotation angle, it is crucial to consider the alteration of the distance between neighboring nanostructures as the rotation angle changes within a fixed period. This change in distance has the potential to influence the near-field coupling condition between the nanostructures (refer to the insets in Fig. 2b), resulting in the observed fluctuations in the reflection intensity. These results validate the geometric phase method for all resonances, even though the multi-resonance property comes from the near-field interaction between the nano-rod structure and the DBR

substrate. Indeed, the selection of the geometric shape for the meta-atom in our study was random and primarily intended for illustrative purposes. The key concept is that other anisotropic nanostructures can be employed to attain similar results and outcomes, as long as their physical dimensions are carefully optimized to maximize the efficiency of circular polarization conversion. To spatially separate a color image into multiple wavelength channels, we design a MOFM by incorporating the multi-resonant meta-atom with transverse chromatic aberration[23,31,32] (see Supplementary Note 1 for more details). It is worth to mention that the multi-resonant meta-atoms produce several near-zero intensity dips that can naturally filter the images. Thus, the crosstalk between wavelength channels is minimized when the MOFM is accordingly optimized and is illuminated under a broadband light source. Figure 2d shows the ray-tracing calculations of the off-axis focusing meta-mirror at four wavelengths using the commercial software OpticStudio (Zemax). In order to capture a complete set of images in a one-shot measurement, it is necessary to ensure that different color channels corresponding to peak wavelengths in Fig. 2a are imaged on a fixed focal plane. Achieving this requires careful consideration of the depth of focus (DOF, which is proportional to the square of focal length) and numerical aperture (NA, which is inversely proportional to the focal length) during the design of the meta-mirror, rather than focusing solely on the focal length. To enable snapshot imaging with acceptable resolution, a focusing meta-mirror with a larger DOF and lower NA is employed. However, this approach involves

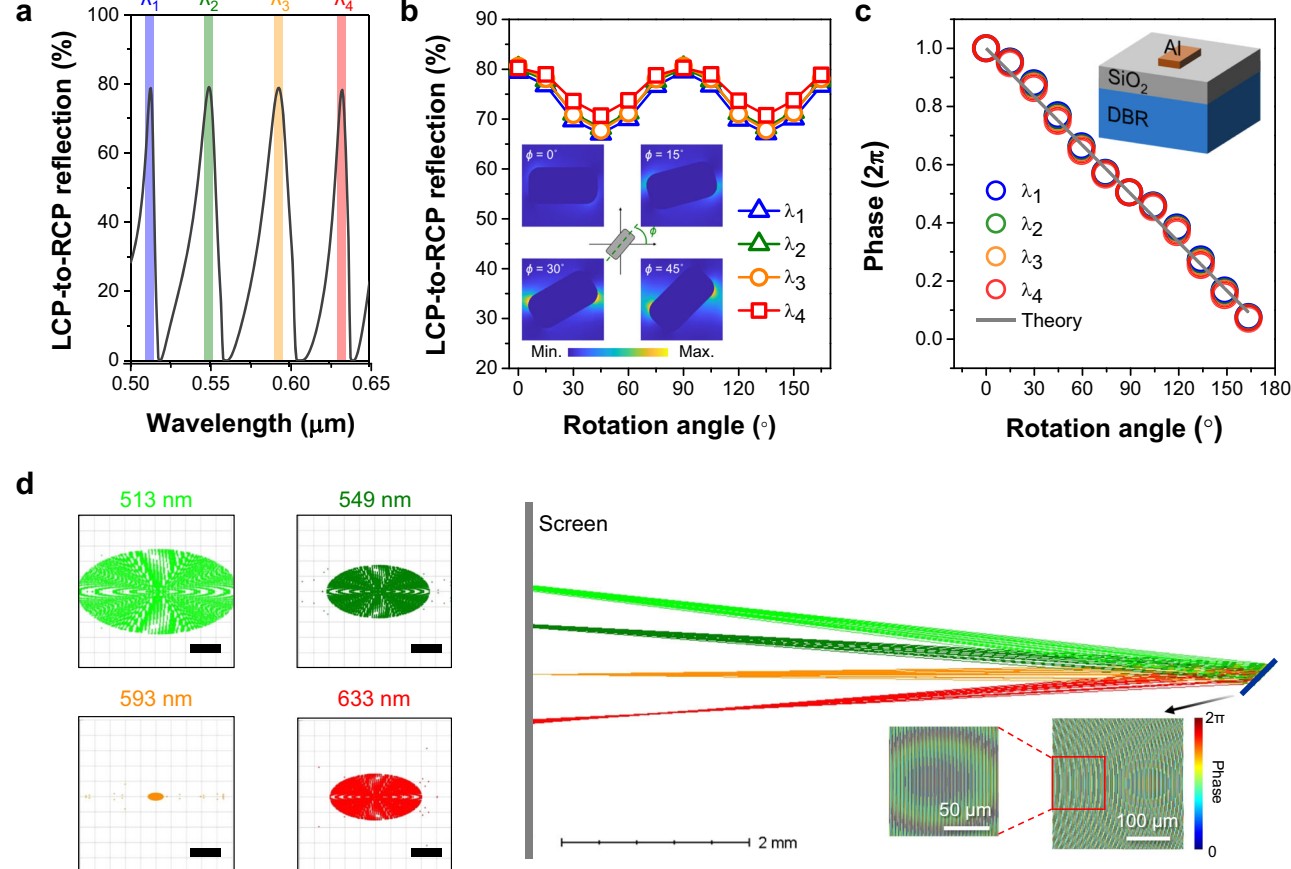

**Fig. 2 | Design of the multi-resonant meta-atom and off-axis focusing meta-mirror. a** The numerical reflection spectrum of the designed multi-resonant meta-atom. An Al nano-rod (length = 170 nm, width = 90 nm, thickness = 50 nm, period = 200 nm) array standing on a DBR substrate is optimized to possess four high-Q resonant peaks across the visible window. The circular cross-polarized reflection (**b**) and phase (**c**) as a function of structural rotation angle, presenting that the designed multi-resonant meta-atom satisfies the geometric phase conditions at

four peak wavelengths. The inset in (**b**) represents the electric field intensity at 593 nm for various structural orientation angles. The inset in (**c**) shoes the schematic of the meta-atom. The thickness of dielectric spacer SiO₂ is 135 nm. **d** Ray-tracing calculations for the off-axis focusing meta-mirror (designed at a central wavelength of 593 nm with a focal length of 7.5 mm) based on the multi-resonant meta-atoms. Left images are the spot diagrams. Scale bars: 10 μm. The bottom right images show the phase distributions across the meta-mirror at 593 nm.

a trade-off, as it may result in reduced image resolution. In our specific case, we have designed a meta-mirror with a NA of 0.02 at a central wavelength of 593 nm to demonstrate snapshot imaging with satisfactory resolution. As shown in Fig. 2d, four focal spots lie on the screen parallel to the dispersion direction. As can be seen, the spot size becomes larger when the incident wavelength is away from the optimized wavelength of 593 nm. Besides the phase deviation, more significant optical aberrations associated with larger focusing angles can also degrade the focusing performance, bringing about the distortion of focal spots at other wavelengths.

### Focusing and imaging performance of multi-wavelength off-axis focusing meta-mirror

Next, we fabricate and experimentally characterize the optical functionality, including the imaging performance of the developed MOFM. We emphasize that the multi-imaging channels are realized using one single metasurface chip, which can significantly improve the working efficiency and reduce the device footprint and fabrication/design complexity. Figure 3a plots the measured focusing efficiency of the fabricated meta-mirror (see Supplementary Fig. 1 for the details of the fabrication process), which shows multiple peaks across the visible window that is highly consistent with the numerical prediction in Fig. 2a. Before demonstrating the snapshot imaging capability, we characterize the imaging performance of the fabricated meta-mirror using single-wavelength lasers as the light source and metallic apertures as the object. The optical configuration is shown in Supplementary Fig. 2a. Figure 3b, c, respectively, show the computed and

experimentally measured images formed by the MOFM at four different wavelengths. As expected, the introduction of transverse optical dispersion assists in the lateral shift between color images. The focusing meta-mirror is particularly designed so that the images of four target wavelengths are certainly separated in space. In addition, it can be found that the number "2" shows the clearest features at the third (optimized) wavelength channel (the yellow image in Fig. 3b). The blurred images at wavelengths away from the central one are again from the optical aberrations caused by the off-axis focusing effect. Indeed, the blurring effect observed in the color images is influenced by both the intrinsic characteristics of the meta-mirror and optical aberrations, which are also correlated with the incident/reflected angle (refer to Supplementary Note 2 for more discussions). Figure 3c shows the measured images, which experimentally verify the transverse optical dispersion effect and the multispectral imaging functionality of the optimized MOFM. Comparison for other metallic apertures can be found in Supplementary Fig. 3, where all experimental results highly match the ray-tracing calculations. Subsequently, we use the optical setup shown in Supplementary Fig. 2b to examine the snapshot imaging capability of the MOFM. A tube lens is used to relay captured images to a visible camera for simplicity. As shown in Fig. 3d, only four images are observed when a color object is imaged. The images other than the central wavelength channel are naturally filtered because of the meta-mirror's multi-wavelength focusing property. Thus, multispectral imaging composed of four wavelength channels is acquired in one measurement. One can see the discrepancy in color at channel 3 between the measured images shown in Fig. 3c, d, which can be

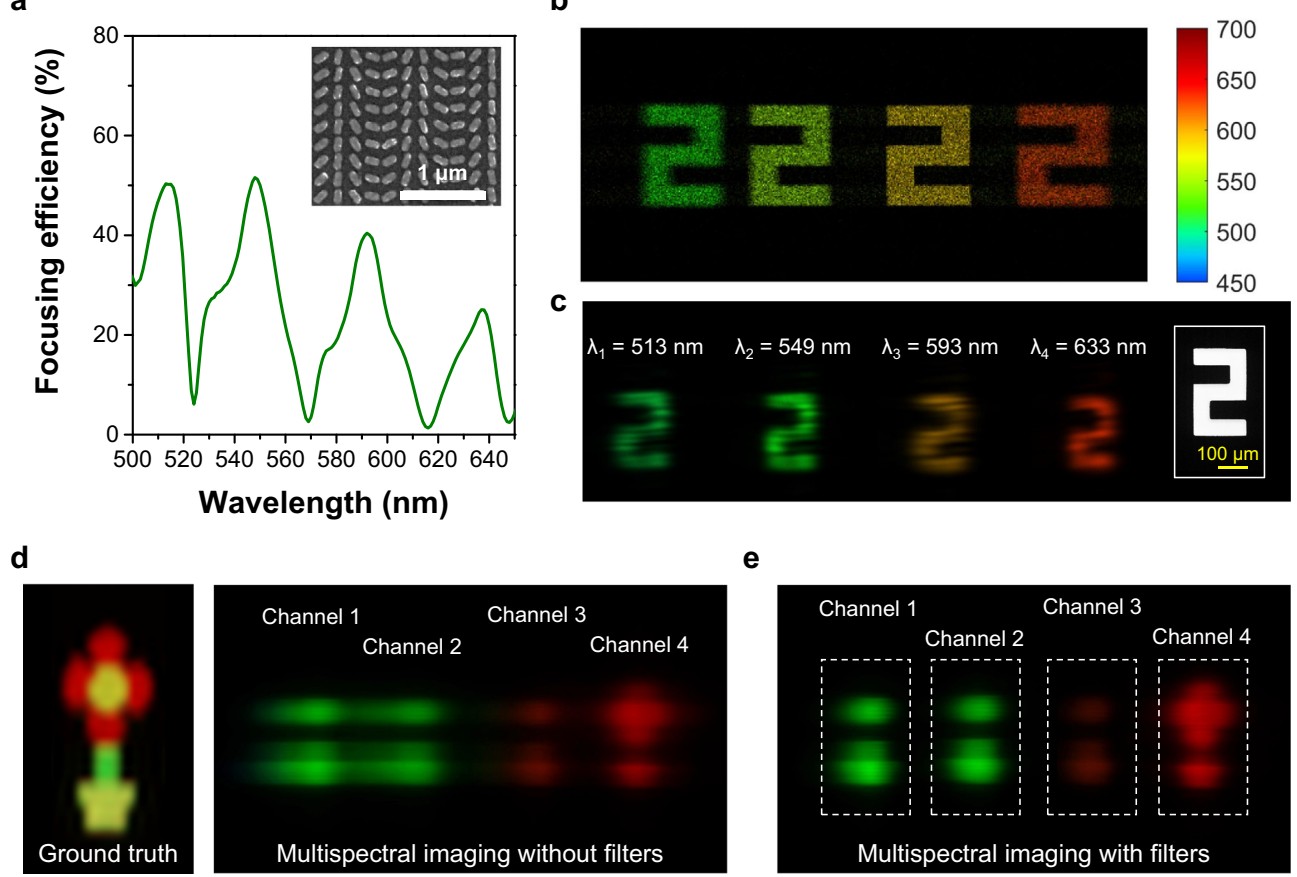

**Fig. 3 | Focusing and imaging demonstration of the multi-wavelength off-axis focusing meta-mirror. a** Measured focusing efficiency as a function of wavelength. The inset shows the SEM image of the fabricated meta-mirror. Calculated (**b**) and experimentally captured (**c**) image of a number "2" at 4 wavelength bands. The right

panel presents the optical microscopic image of the object. Acquired 4-band multispectral images without (**d**) and with (**e**) filters. The ones captured without filters are regarded as a snapshot multispectral dataset. The left panel in (**d**) shows the ground truth.

attributed to the use of different light sources for white balance. However, we emphasize that this variation in color does not impact the accuracy of the multispectral/hyperspectral imaging results. This is because the spatial distribution of wavelength channels ensures that each color image corresponds to its respective wavelength channel at the correct spatial position (refer to Supplementary Note 3 for more discussions). Additionally, we can see that the images at the four channels shown in Fig. 3d are blurrier than those presented in Fig. 3c, which resulted from the relatively low Q-factor of the measured peaks (see Fig. 3a, Supplementary Fig. 2c, d for comparison). The experimentally observed lower Q-factor can be attributed to the imperfection of the fabricated sample. The experimental Q-factor can also be lower when the light source possesses lower coherence[33], which is the case here. The images captured with filters are also shown and compared (refer to Fig. 3e). The position of color filters is optimized to obtain the best-focusing images at target wavelengths. Still, an optical aberration-caused blur exists in the pictures at off-center wavelengths. The color filters enhance the image quality, as shown in the images in Fig. 3e, verifying the above discussions.

## CODE-based small-data learning theory

To obtain a high-quality hyperspectral data cube from the multispectral images, an innovative machine learning theory is developed. Our imaging technique applies the CODE learning theory[34], originally developed for small-data-learning (SMDL) based hyperspectral satellite data restoration (for which data is rare and expensive), where the SMDL is achieved by judiciously blending the advantages of CO and DE to perform the spectral super-resolution to obtain the deblurred hyperspectral image computationally.

As a result, CODE avoids the big data required in DE and the heavy math often encountered in CO. In our application, collecting big data is not economical because of the need for a bunch of color filters. CODE was initially invented for advanced hyperspectral satellite tasks[34] by introducing the $Q$-quadratic norm $\|\cdot\|_Q$ to bridge CO and DE. In this work, the CODE learning theory will be employed for hyperspectral metasurface imaging. We highlight that few data points (only 18 images, each containing just $484 \times 192$ pixels) are sufficient for high-quality hyperspectral imaging reconstruction. Let $Y \in \mathbb{R}^{4 \times L}$ be the meta-mirror-acquired $L$-pixel 4-band multispectral image, from which we aim to reconstruct the corresponding deblurred hyperspectral image $X \in \mathbb{R}^{M \times L}$ (i.e., the target image), where $M$ is the number of hyperspectral bands. Mathematically, $Y$ can be regarded as the blurred low-spectral-resolution counterpart of $X$, and this relation can be concisely modeled as

$$Y = f_B(DX),\qquad(1)$$

where the spectral response matrix performs spectral $D$ downsampling[35], and $f_B(\cdot)$ describes the blurring effect caused by the MOFM. Using the small-data CODE theory, we aim to computationally infer the target image $X$ from the metasurface-acquired information $Y$.

Due to the complicated mechanism of the metasurface system, it is difficult to describe the blurring effect $f_B$ using an explicit mathematical function. Even with the function $f_B$, the naive data-fitting term $\|Y - f_B(DX)\|_F$ is non-convex and demanding to be efficiently addressed, where $\|\cdot\|_F$ is the Frobenius norm. Thus, we propose to learn the inverse blurring procedure $f_B^{-1}(\cdot)$ using the deep Transformer model, as detailed in the CODE Implementation in Supplementary Note 4. Considering that the blurring effect caused by the MOFM is intractable and expected to be highly non-linear, we learn the inverse procedure $f_B^{-1}$ using the customized Transformer. The proposed Transformer is deployed using the U-Net structure, as detailed in Fig. 4, where each Transformer block (T-Block) is also depicted. The T-block revises the Restormer[36] for better interaction among the feature maps. This upgrades the QKV attention effectiveness gained by interchanging the

ordering of the depthwise convolution (Dconv) and the typical convolution block. After the last T-Block that focuses more on spectral attention, we further enhance the spatial features using spatial-spectral domain learning (SDL) module[37], whose output is the desired deblurred multispectral image $\widetilde{Y} = f_B^{-1}(Y)$. It is quite interesting to notice that numerous recent articles have proposed and successfully demonstrated the training of the Transformer with just small data[38–41]. The CODE addresses the challenge of small data learning using a completely different philosophy. Simply speaking, typical techniques[38–41] have to force the deep network to return a good deep solution (as the final solution), while CODE just accepts the weak DE solution. CODE assumes that though the small scale of data results in such a weak solution, the solution itself still contains useful information. Under this assumption, CODE then applies $Q$-norm to extract the embedded useful information to guide the algorithm as a regularizer, thereby yielding the final high-quality solution. We refer interested readers to ref. 34 for an in-depth discussion about the theoretical aspect of why CODE could work very well even in the absence of big data. Please also see Supplementary Note 4 for more discussions.

Next, we explain how to map the multispectral image $\widetilde{Y}$ to the hyperspectral image $X_{DE}$. As previously discussed, the CODE learning theory does not require big data and can accept a roughly estimated solution $X_{DE}$ from small data. For this reason, a simple two-branch convolution neural network (CNN) deployed like Fig. 4 is sufficient to obtain $X_{DE}$ for effectively supporting the subsequent deep regularization to be implemented (refer to Algorithm 1 in Supplementary Note 4).

In our application, the most challenging part lies in increasing the number of spectral bands; specifically, the output of the MOFM has just 4 bands, and we aim to superresolve it to the 18-band hyperspectral image. Thus, we suggest using the color transform (CT), which better captures the nature of hyperspectral images than the rotation transform (RT). Mathematically, given the available small dataset $\mathcal{D} = \{Y_1, Y_2, \ldots, Y_{20}\}$, we can augment it to $\mathcal{D}_{aug} = \{Y_1, Y_1', Y_2, Y_2', \ldots, Y_{20}, Y_{20}'\}$ in which $Y_i'$ is a row-shifted version of $Y_i$. Note that in this work, the rows of the image $Y$ correspond to spectral bands (colors), and columns correspond to pixels; thus, row shift exactly implements the desired CT data augmentation. In practice, our dataset contains 20 pairs of images, each composed of a blurred 4-band multispectral image and a clean 18-band hyperspectral image, both having a spatial size of $484 \times 192$. By utilizing the CT, we augment the dataset to 40 pairs in $\mathcal{D}_{aug}$, of which 36 are for training, 2 for validation, and 2 for testing. As a result, an 18-band high-quality hyperspectral image data cube can be obtained from a 4-band multispectral image. Note that such an imaging task is highly challenging because the algorithm has to return the high-quality hyperspectral image from the optically blurred 4-channel MOFM-acquired data. As per our empirical study, this task cannot be achieved by the typical algorithm (e.g., ADMM) or typical L2-norm regularization. Thus, the CODE theory employs the less-seen $Q$-norm regularization in the ADMM-Adam algorithm. For computational efficiency, the $Q$-norm regularization [see Eq. (S4) in Supplementary Note 4] can be trickily designed based on hyperspectral subspace geometry [see Supplementary Note 4]. Such a tricky geometry design allows us to reformulate the $Q$-norm into the commonly seen L2 norm [see Eq. (S5) in Supplementary Note 4], thereby allowing us to achieve high-speed metasurface-driven hyperspectral imaging. Last, we would like to point out that with all the algorithmic steps solved by closed-form solutions, the imaging algorithm (i.e., Algorithm 1 in Supplementary Note 4) for implementing the CODE-based formulation is high-speed (see more discussions in Supplementary Note 4).

## Snapshot hyperspectral imaging demonstration

Figure 5 a shows the reconstructed 18-band channels of the snapshot hyperspectral imaging dataset for a potted flower from 480 nm to 650 nm. Note that the input images are acquired without color filters, thus,

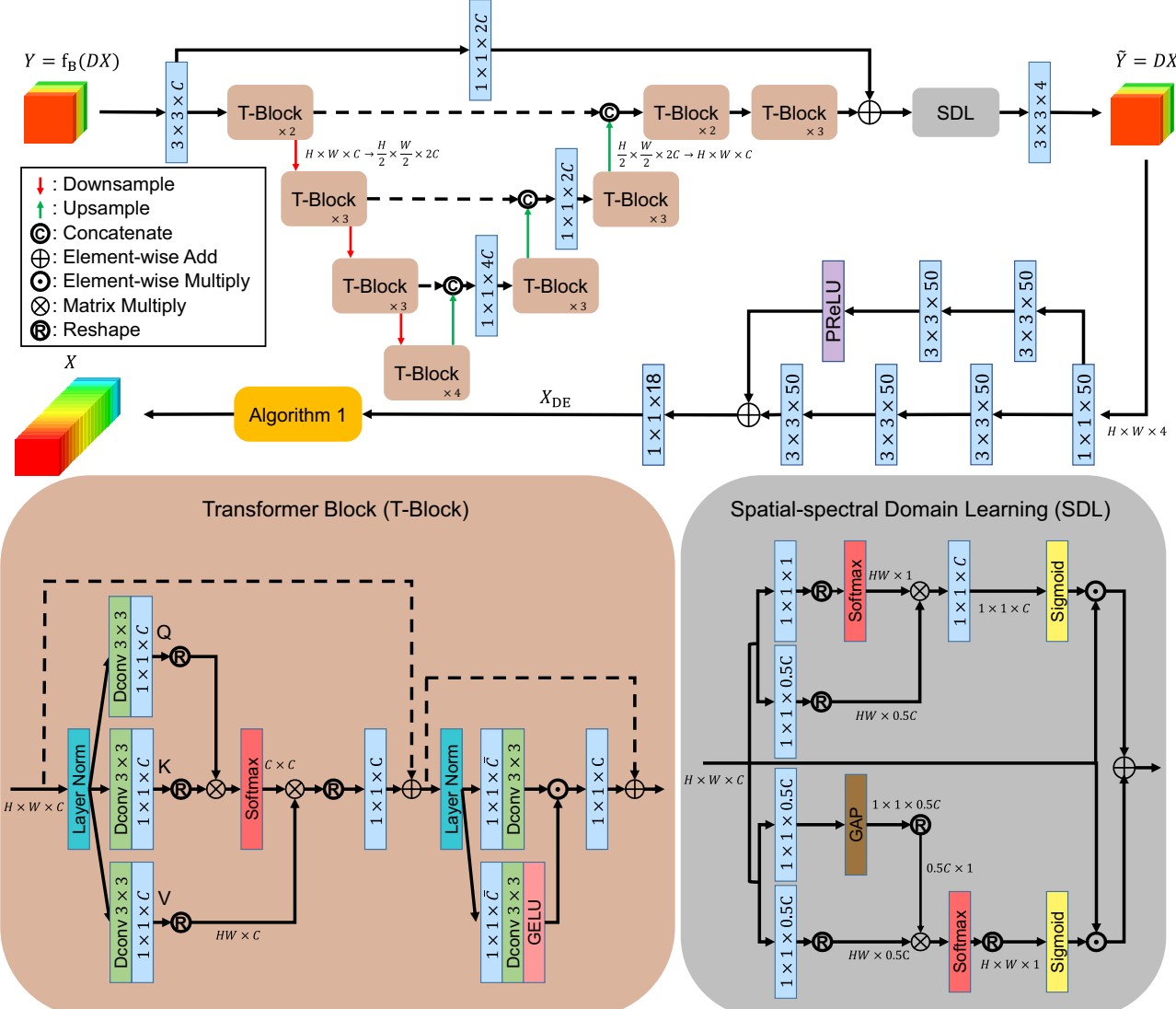

**Fig. 4 | Processing flow of the CODE small-data learning and imaging theory.** The overall design of the CODE learning architecture, where "$k \times k \times c$" denotes a 2D convolution with $c$ kernels sized of $k \times k$, "GAP" means global average pooling, "Layer Norm" performs the commonly seen layer normalization, "Dconv $k \times k$" is the depthwise convolution with a $k \times k$ kernel, "PReLU" denotes the parametric rectified linear unit. The details for Algorithm 1 can be founded in Supplementary Note 4.

a snapshot hyperspectral imager is achieved. Although the input multispectral image is blurred (see Fig. 3d), the generated hyperspectral images are optically clean and close to the ground truth (see Supplementary Fig. 4). A visible color image can be subsequently constructed by assigning the red, green, and blue channels from Fig. 5a. The reconstructed hyperspectral images perfectly retrieve the optical features of the original object, even for the channels that exhibit a relatively weak signal in the color image. For example, the petal shows a high reflection in red color, while the same component in the hyperspectral dataset represents the highest brightness at a 632.8 nm wavelength channel. A weak signal in the flower pot at the cyan color channel (490 nm) is also observed, indicating the robustness of the CODE theory. In Fig. 5c–f, we provide quantitative spectra along the wavelength axis at four pixels of interest. The reconstructed spectra highly match the ground truth at all pixels, even for the wavelength channels missing in the initial multispectral images. Therefore, the fingerprint of the potted flower can be completely recovered and characterized: the spectrum at pixel 1 shows relatively weak intensity at the green channel, leading to the yellow-like color on the stamen; the stem exhibits a green-like color, resulting from the relatively low intensity at long wavelengths in the spectrum (see

Fig. 5e). As a snapshot hyperspectral imager, it is important to acquire all information in one-shot measurement. Although using color filters can improve the image quality of the multispectral dataset (see Fig. 3e), the information of all channels would not be able to be obtained simultaneously in the filter-involved configuration. Indeed, our results verify that the hyperspectral dataset computed from the inputs with the color filters performs nearly identical with those from the images acquired without filters (see Fig. 5, Supplementary Figs. 6 and 7). Another hyperspectral imaging for a hatchet is also provided to demonstrate the high performance of our device, as shown in Supplementary Figs. 5, 8, and 9. These results greatly support our device for snapshot hyperspectral imaging applications.

To quantitatively analyze the imaging performance of the MOFM-based hyperspectral imager, we calculate the pixel-wise mean squared error (MSE) loss which is widely used to compare the MSE at individual pixels from the real image (ground truth) and the generated image. It can be described as:

$$\text{PMSEL}_{\text{band } n} = \sum_{x=1}^{W} \sum_{y=1}^{H} \frac{[I_{\text{image}}(x,y) - I_{\text{GT}}(x,y)]^2}{W \times H} \tag{2}$$

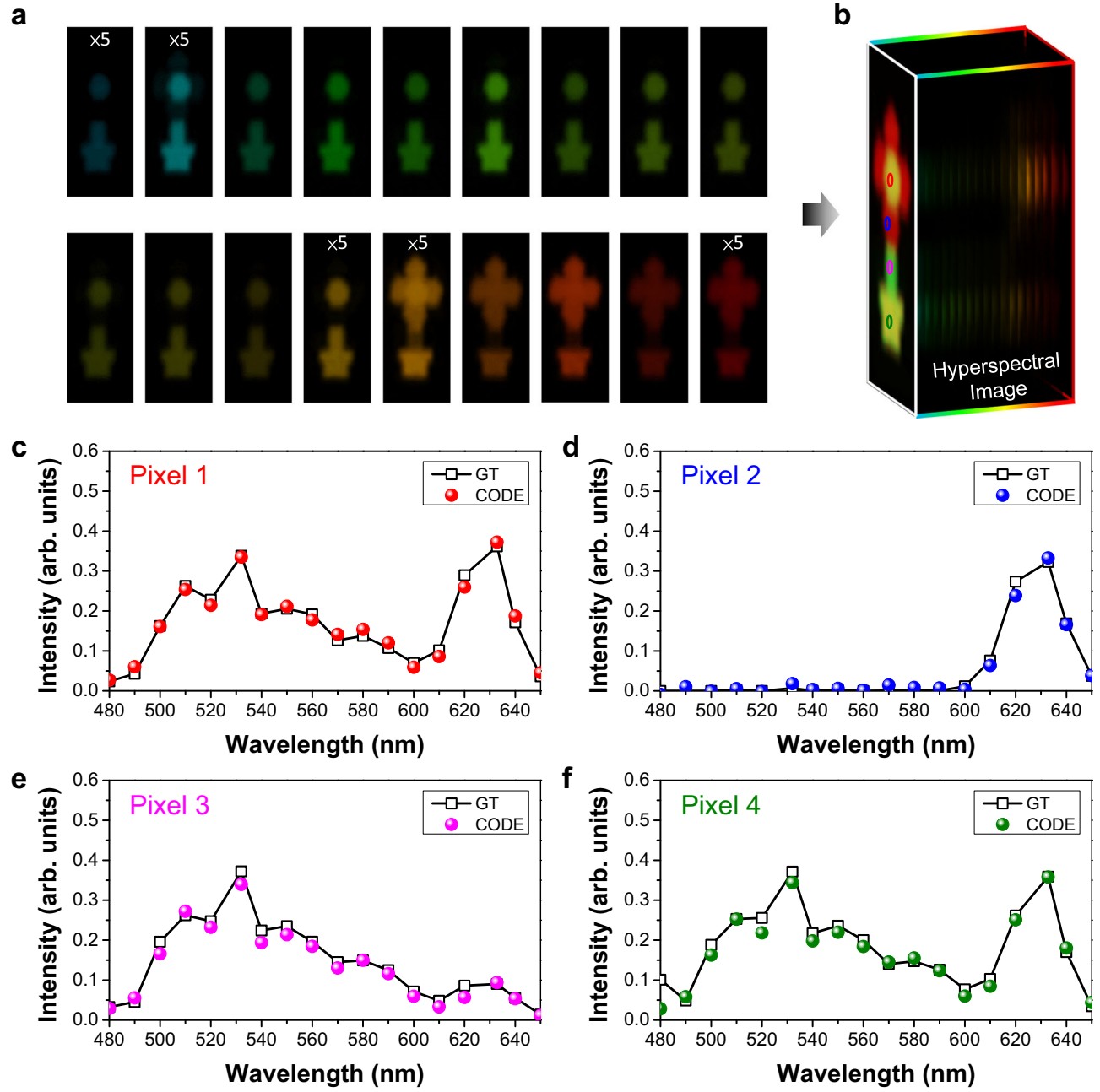

**Fig. 5 | Snapshot hyperspectral imaging. a** Experimentally demonstrated snapshot hyperspectral imaging dataset with a single metasurface chip. The center wavelengths are 480, 490, 500, 510, 520, 532, 540, 550, 560, 570, 580, 590, 600, 610, 620, 632.8, 640, 650 nm. **b** Reconstructed image of the potted flower. **c–f** Spectral data at four pixels highlighted in (b). The ground truth (GT) spectral results are provided for comparison.

where $W$ and $H$ are the total pixels along $x$-direction and $y$-direction, respectively. $I_{image}$ and $I_{GT}$ are the intensity on the reconstructed hyperspectral images and ground truth, respectively. As shown in Fig. 6a, the pixel-wise MSE loss shows shallow values (at an order of $10^{-4}$) across the visible spectrum, indicating the high performance of the developed CODE metasurface-imaging theory for acquiring the hyperspectral data cube in a broad wavelength band. Two objects are imaged for 18-band hyperspectral imaging to demonstrate the versatility of the CODE theory: a potted flower and a hatchet. Additionally, the root-mean-square error (RMSE) and the spectral angle mapper (SAM) of the computed hyperspectral images show low values for two objects, as shown in Fig. 6b, c. The definition of the RMSE and SAM can be found in Supplementary Note 6. Note that SAM is more frequently used in hyperspectral literature as it better

evaluates how useful the hyperspectral data cube is for material identification, well echoing our CT-based SMDL trick. The low angles in SAM spectra directly determine the high spectral similarity between the reconstructed and real hyperspectral images, further proving the high fidelity of our approach. Again, although the input 4-band images are optically blurred, we highlight that all figures of merit calculated from the snapshot hyperspectral imaging dataset (non-filtered multispectral images as input for the CODE technique) are close to those obtained with filters. These results indicate that the proposed approach has a high potential for real-world real-time imaging systems.

## Discussion
This work proposed and experimentally demonstrated a metasurface-empowered snapshot hyperspectral system based on a MOFM in

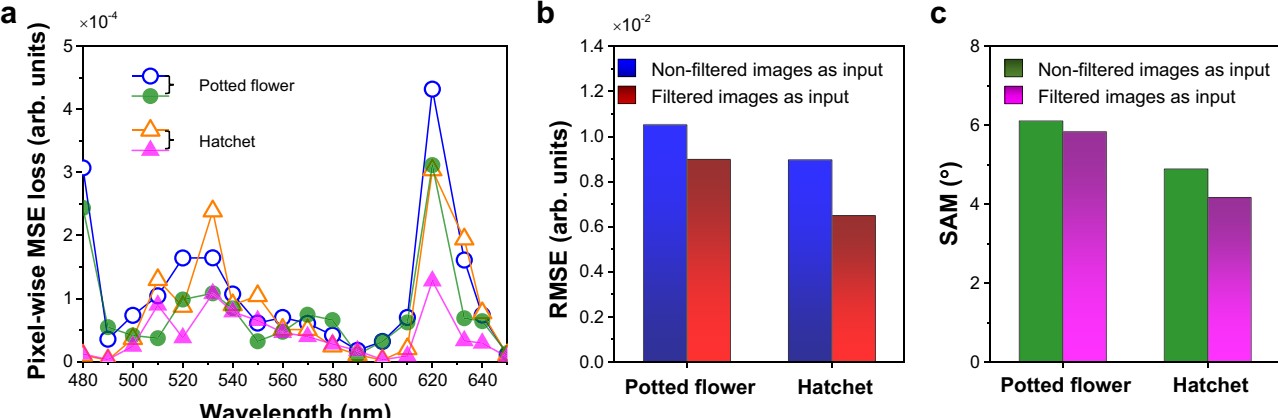

**Fig. 6 | Fidelity of the hyperspectral imaging data cube. a** Calculated pixel-wise mean squared error (MSE) loss versus wavelength of incidence. The hollow(solid) points are calculated from hyperspectral image dataset that are computed from non-filtered(filtered) multispectral images. Root-mean-square error (RMSE) (**b**) and spectral angle mapper (SAM) (**c**) for two hyperspectral imaging data cubes.

conjunction with an innovative CODE-based small-data learning theory. Taking a small dataset with a 4-band image as the input, the system can generate an aberration-corrected hyperspectral data cube composed of 18-bands high-fidelity color images. The initial 4-band multispectral image is acquired using a single metasurface chip, significantly reducing the footprint of the hyperspectral imager. The developed hyperspectral imager's low-profile and small device area makes its high potential for advanced systems and space instruments, such as unmanned aircraft systems and small satellites. While our snapshot hyperspectral system currently demonstrates 18-band data cubes in the visible region, it is worth noting that the number of wavelength bands and the spectral range can be further extended. This is attributed to the multi-resonance feature of the developed metaatom, which covers a wide range from the visible to the near-infrared region (as shown in Supplementary Fig. 11). Theoretically, the working spectral range is naturally constrained by the bandwidth of the DBR's reflection window. In fact, both the number of eigenmodes in the nanostructures and the cavity-like coupling between the topmost nanostructures and the DBR substrate play key roles in determining the number of wavelength channels. By utilizing freeform nanostructures with anisotropic shapes[42,43], the number of eigenmodes can be increased, allowing for a greater number of wavelength bands. Moreover, we investigated the impact of tuning the cavity-like coupling condition on the number of wavelength channels by varying the thickness of the $SiO_2$ spacer, as discussed in Supplementary Note 7. In comparison to the original design with a 135-nm-thick $SiO_2$ spacer, which generated 4 peak wavelengths ranging from 480 nm to 650 nm, the number of wavelength bands increased when $SiO_2$ spacers increases.

As shown in Supplementary Fig. 16, increasing the dielectric spacer thickness to 5000 nm allows for the generation of 12 resonant wavelengths within the same spectral range, which reveals the possibility to further enhance the spectral resolution. To preserve the fidelity of the image throughout the conversion process, we propose adhering to a ratio of 4.5 for the number of bands in the output hyperspectral image compared to the number of bands in the input multispectral image, based on the principles of the CODE theory showcased in this study. Thus, a 12-band multispectral imaging theoretically enables the generation of hyperspectral images with 54 wavelength bands, resulting in an improved wavelength resolution from approximately 10 nm to 3.9 nm. However, it is important to note that this requires providing the training model with ground truth data that corresponds to the specific number of bands. Practical implementation of this approach poses challenges, particularly in obtaining accurate ground truth data through multiple bandpass filters. Therefore, it is crucial to consider the difficulties associated with acquiring precise ground truth data while striving to enhance spectral resolution, and maintain a balanced approach throughout the process. These findings highlight the potential for further extending the number of wavelength bands and spectral range by optimizing the design parameters and cavity-like coupling conditions in future iterations of the system.

Furthermore, we highlight that the CODE theory exhibits the capability to extract the spectral information of an arbitrary pixel A, which possesses a bandwidth of approximately 10 nm, despite the fact that each wavelength channel in the multispectral image contains data with a broader bandwidth of around 20–30 nm (as shown in Fig. 3a). This ability stems from the CODE theory's acquisition of the mapping between the known hyperspectral ground truth and its corresponding input, which is the multispectral image generated by the MOFM. By leveraging this learned mapping, the CODE theory can effectively infer the unknown hyperspectral images that correspond to the multispectral image. Consequently, even if pixel A is exclusively present in a subset or a single spectral band within the ground truth data library, its hyperspectral information can still be accurately deduced through the utilization of the learned mapping from other data pairs.

Another advantage of the CODE learning theory is that only 18 data points are required for the learning process, which is small compared with previous machine learning techniques, dramatically reducing the time and difficulty for data collection. Thanks to the closed-form solutions, the CODE-based learning theory leads to speedy computational time, which highly supports the practical application in high-speed detection. Finally, we point out that the imaging resolution could be further enhanced by more sophisticated deep network architecture or more complicated convex regularization, if the imaging speed is not the main concern. Our metasurface-driven snapshot hyperspectral imager with small-data learning technique can benefit the development of compact spectral imaging devices and their applications in many fields like CubeSat, bio-optical systems, and real-time dynamic detection.

## Methods
### Sample fabrication
First, a 135-nm-thick $SiO_2$ was deposited on a DBR substrate by a magnetron sputtering machine. Then, a 180-nm-thick photoresist (PMMA A4) was spin-coated and baked on a hot plate at 180 °C for 3 min. Afterward, the photoresist was exposed using an electron beam writer (Elionix ELS-7500) at an acceleration voltage of 50 keV with a

beam current of 50 nano-ampere, followed by a development process for 2 mins with the PMMA developer (MIBK: IPA = 1:3). Finally, a 50-nm-thick Al layer was deposited using a thermal evaporator with deposition rate of 1.0 angstrom/s. After the lift-off process using PG remover, the Al nano-rods were defined. See Supplementary Fig. 1 for the schematic of fabrication process.

## Optical setup

The imaging characterization of MOFM was implemented using two optical setups. In the case of using single-wavelength lasers as the light source, metallic apertures with different shapes were employed as the objects (as shown in Supplementary Fig. 2a). In this scenario, a supercontinuum laser (NKT Photonics FIU-15) was combined with an acousto-optic tunable filter (AOTF, SuperK SELECT) to select the desired wavelength within the visible spectrum. A pair consisting of a linear polarizer and a quarter-wave plate was used to control the polarization state of the incident light. On the other hand, to assess the multispectral imaging capability of the MOFM, the laser light source and metallic apertures were substituted with a projector (refer to Supplementary Fig. 2b).

## Data availability

Data underlying the results are available from the corresponding authors upon request.

## Code availability

The code used to generate the hyperspectral imaging dataset is available from the corresponding authors upon request.

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

## Acknowledgements

The authors gratefully acknowledge the use of advanced focused ion beam system (EM025200) of NSTC 112-2731-M-006-001 and electron beam lithography system belonging to the Core Facility Center of National Cheng Kung University (NCKU). The authors acknowledge the support from the National Science and Technology Council (NSTC), Taiwan (Grant number: 111-2112-M-006-022-MY3; 111-2124-M-006-003; 112-2124-M-006-001), and in part from the Higher Education Sprout Project of the Ministry of Education (MOE) to the Headquarters of University Advancement at NCKU. C.-H.L. acknowledges the supports from the Center for Data Science at NCKU, from the EINSTEIN Program of NSTC (Grant number: 111-2636-E-006-028) and from the Emerging Young Scholar Program (i.e., 2030 Program) of NSTC (Grant NSTC 112-2628-E-006-017). P.C.W. also acknowledges the Yushan Fellow Program by the MOE, Taiwan for the financial support. The research is also supported in part by Higher Education Sprout Project, Center for Quantum Frontiers of Research & Technology (QFort) at NCKU. The authors thank Tzu Yuan Lin for useful discussions and help in ray-tracing ZEMAX calculations and Hsuan-Ting Kuo for the assistance in optical measurement.

## Author contributions

C.-H.L. and P.C.W. conceived the original idea. C.-H.L. developed the small-data deep learning theory, derived the convex algorithm with closed-form solutions and convergence guarantee, and prepared the manuscript. S.-H.H. designed and optimized the multi-wavelength meta-atoms and MOFM, built up the optical setup, performed the numerical simulations, prepared the metasurface sample, and performed the optical measurement. T.-H.L. performed the optimization calculations, prepared the calculation code, and implemented the algorithm in MATLAB and Python. P.C.W. organized the project, designed experiments, analyzed the results, and prepared the manuscript. All authors discussed the results and commented on the manuscript.

## Competing interests

The authors declare no competing interests.
