## [Peer Review File · Nature Communications]

Metasurface-empowered Snapshot Hyperspectral Imaging with Convex/Deep (CODE) Small-Data Learning TheoryReviewer #1 (Remarks to the Author):

This paper proposes a computational snapshot hyperspectral imaging system leveraging a meta-lens platform and an optimization-based algorithm for spectral super-resolution. It first designs a multi-wavelength off-axis focusing meta-mirror that employs traverse optical dispersion to focus light of 4 narrow wavelengths into different spots, thus achieving (4 bands) multispectral imaging in one shot. Then, in order to get a high-resolution spectrum, it relies on an optimization algorithm (so called "CODE") to reconstruct 18-band hyperspectral images. An optical setup is built up, and real experiments are conducted to show the effectiveness of the proposed system.

As an expert in signal processing, optimization, and machine learning, my primary focus is to assess the significance of the proposed algorithm. While the optical system and the real imaging results appear to be commendable, I have some reservations about the proposed algorithm. It seems to be a variation of the ADMM algorithm where the objective consists of a data fidelity term along with an L2 norm regularization, which is a commonly employed approach of ADMM in image reconstruction literature.

The authors claim that their method combines convex optimization and deep learning, which they have labeled as CODE small-data learning theory. However, I must admit that I find it somewhat perplexing. I fail to see how this can be referred to as a theory, especially one related to deep learning. The only aspect that could be considered theoretical in the supplementary materials is the convergence analysis of the ADMM algorithm, which is a well-known basic principle in convex optimization. Thus, there doesn't appear to be any truly groundbreaking theory presented.

Furthermore, it strikes me as peculiar that a network is trained using Adam, and its output is then utilized for L2 regularization in ADMM. If we classify this unconventional algorithm as a blending of convex optimization and deep learning, it opens the door to inventing numerous other "algorithms" such as ADMM-SGD, ADMM-AdaDelta, or even ADMM-ADMM (ADMM can be used for training neural networks as well). In fact, I suspect that the positive performance of the proposed algorithm may be a result of overfitting caused by the limited amount of data used for training and testing. Consequently, I find the experimental results presented to be unconvincing and questionable. Perhaps a more direct approach would be to utilize the trained network directly for spectrum recovery. Additionally, have the authors considered replacing the unconventional regularization $\|x - x_{DL}\|$ with a widely-used total variation regularization? This could potentially lead to more reliable and accepted results.

Moreover, the authors state that they employed a transformer-based network for non-linear deblurring, which typically requires a substantial amount of data for effective training. This contrasts with their claim of using small data. Taking all these factors into consideration, I find no substantial grounds for advocating a system that relies on an invalid and ostentatious algorithm.

Reviewer #2 (Remarks to the Author):

The authors propose and experimentally demonstrate a compact snapshot hyperspectral imaging system based on a specifically designed multi-wavelength off-axis focusing meta-mirror (MOFM) and a small-data convex/deep (CODE) deep learning theory. The MOFM enables acquiring a multispectral dataset of 4 images in a one-shot measurement, and the CODE theory can transfer the 4-band multispectral imaging into an 18-band hyperspectral data cube with high fidelity. I think the work is very interesting and the paper is recommended to be published after a few limitations are addressed:

- 1 For the design of the MOFM, how is the transverse chromatic aberration implemented?
- 2 Since the chip works in the reflective mode, what is the impact of the incident angle on the 4-band multispectral imaging?
- 3 What is designed focal length for the working wavelength of 593 nm? If the focal length varies, what is the impact on the 4-band multispectral imaging?
- 4 What are the challenges and possible solutions for increasing the number of wavelength bands and the spectral range? Specifically, the authors mentioned that the multi-resonance feature of the developed meta-atom covers from the visible to the near-infrared region, is it possible to increase

the number of wavelength bands of the initial multispectral image by designing freeform shaped meta-atoms, such as the method used in the visible light band (Laser & Photonics Reviews, 2022: 210066)?

5 For the training of the Transformer network, the authors only used 40 pairs of images and obtained high-quality results. Why does it only need such a small dataset?

Reviewer #3 (Remarks to the Author):

The manuscript presents an experimental demonstration of an 18-channel hyperspectral imaging system. This system utilizes a small form factor of a 4-channel metalens imaging system and employs a small-data convex/deep (CODE) deep learning process. The proposed multi-wavelength off-axis focusing meta-mirror (MOFM) consists of Pancharatnam-Berry (PB) phase Al nanorods and a distributed Bragg reflector (DBR), resulting in multiple channels in reflectance. This characteristic sets it apart from the system described in reference 23. In general, the experimental results align well with the design of the system. The manuscript is well-written and highlights the novel and interesting functionality achieved by incorporating the hot topic of metalens imaging and deep learning in image postprocessing. Overall, with minor revisions addressing the following comments, the manuscript can be accepted for publication.

1. Could the authors provide an explanation for the variation in the circular cross-polarized reflectance spectrum shown in Figure 2b, considering the PB phase scheme where the reflection should be the same under illumination of circularly polarized light?
2. Do the wavelengths of the reflectance resonant peak change when the meta-mirror is illuminated with light at different incidence angles? If yes, there could be a more degree of freedom in terms of wavelength channel in the system.
3. In line 133, the authors mention that the geometric shape of the meta-atom is randomly chosen. However, considering the PB phase scheme, it is expected that the meta-atom should be optimized to achieve maximum circular cross-polarized reflectance at the operation wavelength. Could the authors provide a comment on this apparent contradiction?
4. Could the authors provide an explanation for the discrepancy in color between Figure 3c and Figure 3d, where the image at a wavelength of 593 nm (channel 3) appears yellow in Figure 3c but red in Figure 3d?
5. In lines 184-186, the authors state that the images shown in Figure 3d are blurrier compared to those presented in Figure 3c due to the low Q-factor of the MOFM. Could the authors provide information on the bandwidth of the light beam from the acousto-optic tunable filter (AOTF)?
6. The authors may consider addressing certain limitations in their manuscript. For instance, they can discuss the wavelength resolution of the ground truth, which is mentioned to be 10 nm. In Figure 5a, it is observed that the petal appears in channels ranging from 610 nm to 650 nm, likely due to the broadband nature of the red pixel in the projector. However, what if the red color of the petal is narrowband in nature, such as 10 nm or 20 nm? In such cases, can the CODE generate appropriate 18 channels of hyperspectral images if the petal (or any other color image feature) is only present in a single ground truth channel? It would be valuable for the authors to discuss the limitations of the spectral resolution in their system and explore possibilities for improvement in the discussion section.

In this response letter, we offer a detailed point-by-point reply to each of the reviewers' comments. Our responses are shown *in blue*.

REVIEWERS' COMMENTS:

Reviewer: 1

1. As an expert in signal processing, optimization, and machine learning, my primary focus is to assess the significance of the proposed algorithm. While the optical system and the real imaging results appear to be commendable, I have some reservations about the proposed algorithm. It seems to be a variation of the ADMM algorithm where the objective consists of a data fidelity term along with an L2 norm regularization, which is a commonly employed approach of ADMM in image reconstruction literature.

Reply:

We thank the reviewer for the comment. First, we would like to emphasize that one of the key significances of this work is developing the first algorithm that can successfully construct hyperspectral imaging data cubes using a non-interleaved multi-wavelength metasurface. Such an imaging task is highly challenging because the algorithm has to return a high-quality hyperspectral image from optically blurred 4-channel metasurface-acquired data. As per our empirical study, this task cannot be achieved by the typical algorithm (e.g., ADMM) or typical L2 regularization. Actually, what the CODE theory has employed is the ADMM-Adam algorithm with Q -norm regularization, which is less seen in the literature.

*The L2 norm the referee mentioned comes from a trickily designed Q -norm based on hyperspectral geometry, and such a tricky geometry design allows us to reformulate the Q -norm into the commonly seen (and more easily tractable) L2 norm, thereby allowing us to achieve high-speed metasurface-driven hyperspectral imaging system, which is highly challenging. The detailed reformulation from Q -norm to L2 norm can be found in **Supplementary Note 4**.*

To address the referee's concern and clarify the above points, we added the following discussions into the main text and Supplementary Material:

On Page 11 of the main text

*“...Note that such an imaging task is highly challenging because the algorithm has to return the high-quality hyperspectral image from the optically blurred 4-channel MOFM-acquired data. As per our empirical study, this task cannot be achieved by the typical algorithm (e.g., ADMM) or typical L2-norm regularization. Thus, the CODE theory employs the less-seen Q -norm regularization in the ADMM-Adam algorithm. For computational efficiency, the Q -norm regularization [see Eq. (S4) in **Supplementary Note 4**] can be trickily designed based on hyperspectral subspace geometry [see **Supplementary Note 4**]. Such a tricky geometry design allows us to reformulate the Q -norm into the commonly seen L2 norm [see Eq. (S5) in **Supplementary Note 4**], thereby allowing us to achieve high-speed metasurface-driven hyperspectral imaging...”*

In Supplementary Note 4

“...It is known that pixels of a hyperspectral image often concentrate on a hyperspectral simplex (i.e., convex hull of the affinely independent hyperspectral signatures of the underlying materials in a given image). This hyperspectral geometry then implies that those hyperspectral pixels often distributed in a low-dimensional hyperspectral subspace⁸, motivating us to design \mathbf{Q} using the basis vectors of the subspace. Let $\mathbf{E} \in \mathbb{R}^{M \times N}$ be an orthogonal basis of the hyperspectral subspace, which can be obtained by applying principal components analysis to the roughly estimated imaging result \mathbf{X}_{DE} . Then, we have $\mathbf{X}_{\text{DE}} = \mathbf{E}\mathbf{S}_{\text{DE}}$ for some eigen-image⁴ $\mathbf{S}_{\text{DE}} \in \mathbb{R}^{N \times L}$, whose number of parameters is NL , which is much fewer than that of the target image $\mathbf{X} = \mathbf{E}\mathbf{S} \in \mathbb{R}^{M \times L}$ in Eq. (S3) and leads to a faster algorithm. Therefore, a trickily designed geometry-driven PSD matrix \mathbf{Q} allows us to reformulate the \mathbf{Q} -norm into the commonly seen (and more easily tractable) L2 norm. To be mathematically rigorous, let $\text{vec}(\cdot)$ be the vectorization operator and $\text{vec}^{-1}(\cdot)$ be the inverse operator, and select the geometry-driven $\mathbf{Q} = \mathbf{I}_L \otimes (\mathbf{E}\mathbf{E}^T)$ to build the relation that $\|\mathbf{x} - \mathbf{x}_{\text{DE}}\|_{\mathbf{Q}}^2$ is proportional to $\|\mathbf{s} - \mathbf{s}_{\text{DE}}\|_2^2$, which allows us to concisely reformulate Eq. (S3) into the eigenspace as the convex problem:...”

2. The authors claim that their method combines convex optimization and deep learning, which they have labeled as CODE small-data learning theory. However, I must admit that I find it somewhat perplexing. I fail to see how this can be referred to as a theory, especially one related to deep learning. The only aspect that could be considered theoretical in the supplementary materials is the convergence analysis of the ADMM algorithm, which is a well-known basic principle in convex optimization. Thus, there doesn't appear to be any truly groundbreaking theory presented. Furthermore, it strikes me as peculiar that a network is trained using Adam, and its output is then utilized for L2 regularization in ADMM. If we classify this unconventional algorithm as a blending of convex optimization and deep learning, it opens the door to inventing numerous other "algorithms" such as ADMM-SGD, ADMM-AdaDelta, or even ADMM-ADMM (ADMM can be used for training neural networks as well). In fact, I suspect that the positive performance of the proposed algorithm may be a result of overfitting caused by the limited amount of data used for training and testing. Consequently, I find the experimental results presented to be unconvincing and questionable. Perhaps a more direct approach would be to utilize the trained network directly for spectrum recovery. Additionally, have the authors considered replacing the unconventional regularization $\|\mathbf{x} - \mathbf{x}_{\text{DL}}\|$ with a widely-used total variation regularization? This could potentially lead to more reliable and accepted results.

Reply:

We thank the reviewer for the comments. However, we would like to clarify that in this work, the theory being discussed does not emphasize on the convergence. Also, our convergence analysis is actually not the well-known basic one from Professor Stephen Boyd's seminal paper (Found. Trends Mach. Learn. 3, 1-122 (2011)), but rather a customized convergence condition that is

less commonly seen, which we presented in our published convex optimization book (refer to Ref. 5 in Supplementary Material). Furthermore, what we emphasize through the manuscript is the small-data learning theory (rather than convergence theory), which has been successfully achieved by cleverly blending convex optimization (CO) and deep learning (DE). Theoretically, the effectiveness of Q -norm (used to bridge CO and DE) can be partially attributed to its more generalized form compared to other typical regularization like Tikhonov.

Indeed, the line of blending CO and DE has emerged as a recent trend in addressing hardly-tractable challenging imaging inverse problems (e.g., metasurface-engaged hyperspectral imaging). Besides our CODE, another remarkable technique in this line of research is deep plug-and-play (deep PnP). However, it is important to highlight a key distinction between CODE and deep PnP. While CODE leverages small data for its implementation, the deep PnP technique requires big data to train the deep proximal operator in a convex algorithm. This distinction makes CODE groundbreaking in its ability to achieve high-quality imaging results with limited data. In fact, using only CO or DE individually cannot achieve such a high-quality imaging result from the absence of big data, as recently demonstrated in the first CODE paper (see *IEEE Trans. Geosci. Remote Sens.* **60**, 1-16 (2022)). Merely using DE yields rather weak solution due to the small scale of the available data, but CO (the convex Q -norm regularization, in particular) can leverage such a weak solution to regularize the final solution thereby achieving high-quality hyperspectral imaging results.

Finally, we agree with the referee that ADMM (resp., ADAM) can be replaced by any other suitable CO optimizers (resp., DE optimizers). This is precisely why we replaced “ADMM-Adam” with “CODE” in our recent works. For example, the algorithm for hyperspectral satellite change detection (HCD) has been named CODE-HCD rather than ADMM-ADAM-HCD (refer to *IEEE Trans. Geosci. Remote Sens.* **61**, 1-18 (2023)), where the detection result obtained by only using DE was also proven to be relatively weak. To provide further support for our statements, we added new calculation results to illustrate the value of CODE, by showing that directly using only CO or DE cannot achieve good spectrum recovery imaging results (see newly added **Supplementary Table 2**). In order to use CO alone, we need to remove the DE-based Q -norm regularization. For this new demonstration, we replaced Q -norm with the CO-based total variation (TV) regularization, as the referee suggested. However, the much more complicated form of TV leads to significantly longer computational time and weaker spectrum recovery compared to the mathematically simple form of Q -norm.

As a side remark, though very recently CODE has achieved numerous challenging hyperspectral signal processing tasks, including the tensor completion (i.e., image inpainting) and satellite change detection, the naïve version of CODE is far from being applicable to the hardly tractable hyperspectral metasurface imaging, for which we further introduced the deep data-fitting (besides the deep regularization in the naïve CODE) via Transformer; see **Supplementary Note 4** and **Supplementary Figure 15** for detailed illustration. In this work, we have applied the CODE framework to metasurface-engaged hyperspectral imaging for the first time. Therefore, we respectfully invite the referee to consider the significance of the CODE framework in enabling successful hyperspectral imaging with optically-blurred multispectral

imaging inputs from a non-interleaved multi-wavelength metasurface, while recognizing that the specific choice of optimizer may not be the primary determining factor in the groundbreaking achievements of the proposed algorithm.

To address the referee’s concern and clarify the above points, we added the following results and discussions into the Supplementary Material:

“**Supplementary Table 2.** Quantitative comparison among the proposed CODE, the pure CO imaging technique (i.e., CO-TV) and the pure DE imaging techniques (i.e., DE-1 and DE-2), where TV refers to total variation. DE-1 and DE-2 are also known as MST++ and HRNet, respectively. The smaller the value of RMSE and SAM, the better the reconstruction fidelity (boldfaced number indicates the best performance). See **Supplementary Note 4** and **Supplementary Note 5** for more details.”

Data	Methods	RMSE	SAM	Time (sec.)
Potted Flower (Non-filtered)	CODE	0.01051	6.110	3.94
	CO-TV	0.03821	49.955	163.15
	DE-1	0.02334	9.819	1.01
	DE-2	0.02632	9.085	3.55
Hatchet (Non-filtered)	CODE	0.00896	4.889	3.72
	CO-TV	0.02228	50.352	171.26
	DE-1	0.02046	9.166	1.35
	DE-2	0.02072	5.361	3.72
Potted Flower (Filtered)	CODE	0.00898	5.831	4.04
	CO-TV	0.03814	49.912	160.12
	DE-1	0.02646	11.662	1.02
	DE-2	0.02661	10.227	3.91
Hatchet (Filtered)	CODE	0.00649	4.167	4.01
	CO-TV	0.02215	50.287	166.30
	DE-1	0.02157	9.675	1.37
	DE-2	0.01895	6.554	4.05

In Supplementary Note 4

“..As a side remark, though CODE has achieved numerous challenging hyperspectral signal processing tasks in very recent literature, including the tensor completion (i.e., image inpainting)³ and satellite change detection¹², the naïve version of CODE is far from being applicable to the hardly tractable hyperspectral metasurface imaging, for which we further introduced the deep data-fitting (besides the deep regularization in the naïve CODE); this can be graphically illustrated using the notation system built above, as displayed in the **Supplementary Figure 15.**”

(a) Newly designed CODE for hyperspectral metasurface imaging.

(b) Naïve CODE proposed in Ref. 3.

Supplementary Figure 15. The naïve CODE (bottom panel) involves only deep Q-norm regularization. To address the hardly tractable hyperspectral metasurface imaging, another Transformer-based deep data-fitting term has been further introduced into the newly designed CODE (top panel).

“...1. [Remark 1] Although the standard ADMM convergence conditions (see, e.g., Ref. 9) are not straightforwardly seen, the tricky reformulation of Eq. (S7) ensures the full rankness of the linear association between the two primal variables. According to Ref. 5, such a full rankness can directly build the desired convergence guarantee, as theoretically guaranteed as follows:...”

“...3. [Remark 3] One may wonder how such a mathematically simple CODE theory could be so powerful, but by noticing that the Q-norm function (used to bridge CO and DE) is actually a more generalized version of the classical Tikhonov regularization function³, the good performance becomes not so surprising. Note also that the line of blending CO and DE has become a recent trend in solving hardly-tractable challenging imaging inverse problems (e.g., metasurface hyperspectral imaging). Besides our CODE theory, the most remarkable technique in this line is deep plug-and-play (deep PnP)^{10, 11}. However, unlike CODE that just needs small data, the deep PnP technique requires big data to implement the deep proximal operator in the convex ADMM algorithm^{10, 11}. With this regard, the CODE theory is quite appealing and promising for other challenging imaging applications.”

“...4. [Remark 4] For hardly tractable inverse problems (e.g., restoration of damaged hyperspectral satellite data), merely using either CO or DE is far from being sufficient especially in the absence of big data, as recently demonstrated in the first CODE paper³. Merely using DE yields rather weak solution due to the small scale of the available data, but CO (the convex Q-norm regularization, in particular) can employ such a weak solution to regularize the final solution thereby obtaining high-quality hyperspectral imaging results. ADMM (resp., ADAM) is

not the key and can be replaced by any other suitable CO optimizers (resp., DE optimizers). That's exactly why "ADMM-Adam" has been renamed "CODE" in recent works. For example, the algorithm for hyperspectral satellite change detection (HCD) has been named as CODE-HCD¹², where the detection result obtained by merely using DE was also proven to be rather weak. The value of the CODE framework can indeed be effectively highlighted by comparing the imaging results obtained from CODE with those from other approaches, as can be seen in **Supplementary Table 2**. We added some new results to illustrate the value of CODE, by showing that directly using only CO or only DE cannot achieve good spectrum recovery imaging. In order to use DE alone, we employ two state-of-the-art spectrum recovery DE techniques, known as MST++ (i.e., DE-1)¹³ and HRNet (i.e., DE-2)¹⁴, both of which are trained using exactly the same small data as CODE. To directly use CO only, we need to remove the DE-based Q-norm regularization; thus, for this demonstration, we replaced Q-norm by the prestigious CO-based total variation (TV) regularization. The resulting algorithm, named CO-TV, is derived and detailed in **Supplementary Note 5**. The results are summarized in the **Supplementary Table 2**. One can see that results gained by using only CO or DE individually (i.e., CO-TV, DE-1 and DE-2) are far from being satisfactory for both the filtered and non-filtered cases. Also, the much more complicated form of TV leads to significantly longer computational time, compared to the Q-norm of mathematically simple form."

with six additional references

9. Boyd, S., Parikh, N., Chu, E., Peleato, B., Eckstein, J. Distributed optimization and statistical learning via the alternating direction method of multipliers. *Found. Trends Mach. Learn.* **3**, 1-122 (2011).
10. Fu, X., Jia, S., Zhuang, L., Xu, M., Zhou, J., Li, Q. Hyperspectral anomaly detection via deep plug-and-play denoising CNN regularization. *IEEE Trans. Geosci. Remote Sens.* **59**, 9553-9568 (2021).
11. Dian, R., Li, S., Kang, X. Regularizing Hyperspectral and multispectral image fusion by CNN denoiser. *IEEE Trans. Neural Netw. Learn. Syst.* **32**, 1124-1135 (2021).
12. Lin, T. H., Lin, C. H. Hyperspectral change detection using semi-supervised graph neural network and convex deep learning. *IEEE Trans. Geosci. Remote Sens.* **61**, 1-18 (2023).
13. Cai, Y., Lin, J., Lin, Z., Wang, H., Zhang, Y., Pfister, H., *et al.* MST++: Multi-stage spectral-wise transformer for efficient spectral reconstruction. *2022 IEEE/CVF Conference on Computer Vision and Pattern Recognition Workshops (CVPRW)*; 2022. pp. 744-754.
14. Zhao, Y., Po, L.-M., Yan, Q., Liu, W., Lin, T. Hierarchical regression network for spectral reconstruction from rgb images. *2020 IEEE/CVF Conference on Computer Vision and Pattern Recognition Workshops (CVPRW)*; 2020. pp. 422-423.

We also added the following discussions into the Supplementary Material to describe how we implement the CO-TV computational imaging algorithm for comparison:

"Supplementary Note 5: CO-TV computational imaging algorithm

As discussed in **Supplementary Note 4**, we replaced the DE-based Q-norm regularization by the CO-based TV regularization, and the resulting CO-TV algorithm will be derived below. Specifically, the new imaging criterion becomes

$$\mathbf{X}^* = \arg \min_{\mathbf{X}} \|\tilde{\mathbf{Y}} - \mathbf{D}\mathbf{X}\|_F^2 + \lambda \text{TV}(\mathbf{X}), \quad (\text{S8})$$

where the data-fitting term remains the same, while the regularization term becomes $\text{TV}(\mathbf{X})$ (i.e., the prestigious total variation function)¹⁵. Then, let us reformulate Eq. (S8) into the standard ADMM-form as

$$\min_{z=x} \|\tilde{\mathbf{y}} - (\mathbf{I}_L \otimes \mathbf{D})\mathbf{x}\|_2^2 + \lambda \text{TV}(z), \quad (\text{S9})$$

where \otimes is the Kronecker product, $\mathbf{x} = \text{vec}(\mathbf{X})$, and $\tilde{\mathbf{y}} = \text{vec}(\tilde{\mathbf{Y}})$. We can then solve Eq. (S9) through the ADMM algorithm, as detailed in **Algorithm 2** given below, where the augmented Lagrangian is defined as $\mathcal{L}(\mathbf{x}, z, \mathbf{d}) = \|\tilde{\mathbf{y}} - (\mathbf{I}_L \otimes \mathbf{D})\mathbf{x}\|_2^2 + \lambda \text{TV}(z) + \frac{\mu}{2} \|\mathbf{x} - z - \mathbf{d}\|_2^2$ with $\mu > 0$ being the penalty parameter, q is the iteration number, $\mathbf{d} \in \mathbb{R}^{ML}$ is the scaled dual variable, and the matrix $\mathbf{P} \triangleq (\mathbf{I}_L \otimes \mathbf{D})$. Note that Line 3 in **Algorithm 2** is nothing but the TV denoising operator with $(\mathbf{x}^q - \mathbf{d}^q)$ being the noisy input image. So, the denoising operator of Line 3 can be implemented by using the split Bregman method, whose implementation is freely available online¹⁶. Thus, the derivation of the CO-TV hyperspectral imaging algorithm has been completed.”

Algorithm 2 The CO-TV Hyperspectral Imaging Algorithm.

- 1: **Given** $\lambda := 0.1$, $\mu := 10$. Initialize $\mathbf{x}^0 := \mathbf{0}$, $\mathbf{d}^0 := \mathbf{0}$, and $q := 0$.
 - 2: **repeat**
 - 3: Update $z^{q+1} \triangleq \arg \min_z \mathcal{L}(\mathbf{x}^q, z, \mathbf{d}^q) = \frac{1}{2} \|(\mathbf{x}^q - \mathbf{d}^q) - z\|_2^2 + \frac{\lambda}{\mu} \text{TV}(z)$;
 - 4: Update $\mathbf{x}^{q+1} \triangleq \arg \min_{\mathbf{x}} \mathcal{L}(\mathbf{x}, z^{q+1}, \mathbf{d}^q) := (2\mathbf{P}^T\mathbf{P} + \mu\mathbf{I}_L)^{-1}(2\mathbf{P}^T\tilde{\mathbf{y}} + \mu(z^{q+1} + \mathbf{d}^q))$;
 - 5: Update $\mathbf{d}^{q+1} := \mathbf{d}^q + z^{q+1} - \mathbf{x}^{q+1}$;
 - 6: $q := q + 1$;
 - 7: **until** the predefined stopping criterion is met.
 - 8: Compute $\mathbf{x}^* := \mathbf{x}^q$ and $\mathbf{X}^* := \text{vec}_{M \times L}^{-1}(\mathbf{x}^*)$.
 - 9: **Output** the imaging result \mathbf{X}^* .
-

with two additional references

15. Osher, S., Burger, M., Goldfarb, D., Xu, J., Yin, W. An iterative regularization method for total variation-based image restoration. *Multiscale Model. Simul.* **4**, 460-489 (2005).
16. Benjamin Tremoulheac (2023). Split Bregman method for total variation denoising (<https://www.mathworks.com/matlabcentral/fileexchange/36278-split-bregman-method-for-total-variation-denoising>), MATLAB Central File Exchange. Retrieved July 9, 2023.

3. Moreover, the authors state that they employed a transformer-based network for non-linear deblurring, which typically requires a substantial amount of data for effective training. This contrasts with their claim of using small data.

Reply:

We thank the reviewer for the truly expert comment. However, we respectfully comment that numerous articles have proposed and successfully demonstrated the training of the Transformer with just small data. Some samples are listed below for your information (refer to newly added Refs. 38-41 in the main article). Thus, we do not think there is any contradiction.

Remarkably, CODE addresses the challenge of small data learning using a completely different philosophy. Simply speaking, typical techniques like discussions in newly added Refs. 38-41 in the main article have to force the network to return a good DE solution, while CODE just accepts the weak DE solution (but extracting useful information embedded in such a weak solution to regularize the final solution). By accepting the weak DE solution and utilizing it in the regularization process, CODE leverages the complementary strengths of CO and DE, leading to improved performance and robustness in small data scenarios.

To address the referee's concern and clarify the above points, we added the following discussions into the main article:

On Page 10

*“...It is quite interesting to notice that numerous recent articles have proposed and successfully demonstrated the training of the Transformer with just small data³⁸⁻⁴¹. Remarkably, CODE addresses the challenge of small data learning using a completely different philosophy. Simply speaking, typical techniques³⁸⁻⁴¹ have to force the deep network to return a good deep solution (as the final solution), while CODE just accepts the weak DE solution. CODE assumes that though the small scale of data results in such a weak solution, the solution itself still contains useful information. Under this assumption, CODE then applies Q-norm to extract the embedded useful information to guide the algorithm as a regularizer, thereby yielding the final high-quality solution. We refer interested readers to Ref. [34] for an in-depth discussion about the theoretical aspect of why CODE could work very well even in the absence of big data. Please also see **Supplementary Note 4** for more discussions...”*

with four additional references

38. Shao, R., Bi, X. J. Transformers meet small datasets. *IEEE Access* **10**, 118454-118464 (2022).
39. Lee, S. H., Lee, S., Song, B. C. Vision transformer for small-size datasets. 2021. arXiv:2112.13492.
40. Liu, Y., Sangineto, E., Bi, W., Sebe, N., Lepri, B., De Nadai M. Efficient training of visual transformers with small datasets. *Advances in Neural Information Processing Systems* **34**, 23818-23830.
41. Cao, Y.-H., Yu, H., Wu, J. Training vision transformers with only 2040 images. *Computer Vision – ECCV 2022: 17th European Conference, Tel Aviv, Israel, October 23–27, 2022, Proceedings, Part XXV*. Tel Aviv, Israel: Springer-Verlag; 2022. pp. 220–237.

Reviewer: 2

The authors propose and experimentally demonstrate a compact snapshot hyperspectral imaging system based on a specifically designed multi-wavelength off-axis focusing meta-mirror (MOFM) and a small-data convex/deep (CODE) deep learning theory. The MOFM enables acquiring a multispectral dataset of 4 images in a one-shot measurement, and the CODE theory can transfer the 4-band multispectral imaging into an 18-band hyperspectral data cube with high fidelity. I think the work is very interesting and the paper is recommended to be published after a few limitations are addressed.

We are thankful to the reviewer for the positive evaluation of our work. We carefully addressed the points brought up by the reviewer below.

1. For the design of the MOFM, how is the transverse chromatic aberration implemented?

Reply:

*We thank the reviewer for bringing up this point. In the case of an on-axis dispersive meta-mirror, the focal points for different wavelengths are located at different focal planes due to the inherent optical dispersion. This characteristic, referred to as longitudinal chromatic aberration, poses a challenge for capturing color images in a single-shot measurement as the multispectral images at different wavelength channels would appear at different focal planes (see newly added **Supplementary Figure 12a**). In contrast, the off-axis focusing meta-mirror allows for the appearance of different color channels with a lateral shift, referred to as transverse chromatic aberration. This feature enables the multispectral imaging of a color object onto a tilted focal plane, as illustrated in the newly added **Supplementary Figure 12b**. Therefore, by utilizing the off-axis focusing configuration, transverse chromatic aberration can be effectively introduced into the meta-mirror, facilitating the capture of color images in a single-shot measurement. Furthermore, the multi-wavelength resonant feature of the meta-atoms employed in the meta-mirror aids in minimizing cross-talk between each color channel. This is beneficial for achieving accurate and reliable multispectral imaging, where the different colors can be accurately separated and distinguished.*

To address this issue, we revised the discussion in the main text:

On Page 6 of the main text

“...To spatially separate a color image into multiple wavelength channels, we design a MOFM by incorporating the multi-resonant meta-atom with transverse chromatic aberration^{23, 31, 32} (see **Supplementary Note 1 for more details)...”**

We also added a paragraph and a figure to the Supplementary Material.

Supplementary Note 1: Implementation of transverse chromatic aberration

“In the case of a regular metalens or a focusing meta-mirror, which exhibits longitudinal chromatic aberration, capturing color images in a single-shot measurement is not suitable due to the multispectral images appearing at different focal planes for different wavelength channels (refer to **Supplementary Figure 12a). To overcome this limitation, we propose incorporating transverse chromatic aberration into the focusing meta-mirror. To achieve this, a multi-wavelength meta-mirror is specifically designed with an off-axis focusing effect. By introducing**

transverse chromatic aberration, we can enable the capture of color images in a single-shot measurement, improving the imaging capabilities of the meta-mirror. Building upon previous research conducted¹, the phase distribution of an off-axis meta-mirror/metalens can be simplified as follows:

$$\Phi_{metalens}(r_p, \varphi_p) = \frac{2\pi}{\lambda_d} (f - \sqrt{f^2 + r_p^2 - 2r_p f \sin \theta_f \cos \varphi_p}) \quad (S1)$$

where λ_d is the designed central wavelength, f is the focal length for λ_d , and θ_f represents polar angle. r_p represents the distance between the meta-mirror center and an arbitrary position on the meta-mirror surface. φ_p denotes the angle between the x -axis and the line connecting the center of the meta-mirror to the arbitrary position on the meta-mirror surface. In our specific case, the focusing meta-mirror is designed to operate at a central wavelength of 593 nm with a focal length of 7.5 mm. This design enables the formation of color images with different wavelength channels that are away from the central wavelength. As a result, these color images appear on the screen along the dispersion direction, as shown in **Supplementary Figure 12b**. This characteristic is advantageous for the development of snapshot multispectral imaging, where multiple wavelength channels can be captured simultaneously in a single shot.”

Supplementary Figure 12. Schematic illustration of the chromatic meta-mirror. a On-axis dispersive meta-mirror. **b** Off-axis dispersive meta-mirror. The θ_f is designed at 45° for 593 nm.

with an additional reference

1. Chen, B. H., Wu, P. C., Su, V.-C., Lai, Y.-C., Chu, C. H., Lee, I. C., *et al.* GaN metalens for pixel-level full-color routing at visible light. *Nano Lett.* **17**, 6345-6352 (2017).

2. Since the chip works in the reflective mode, what is the impact of the incident angle on the 4-band multispectral imaging?

Reply:

*We thank the reviewer for pointing this out. The angle of incidence indeed influences the reflected angles for each wavelength band, thereby affecting the lateral separation between different color channels. Moreover, variations in the incident angle have a notable impact on the optical coma aberration, which can significantly affect image quality. As a result, changes in the incident angle can also impact the image quality of individual images. To further elucidate the influence of the incident angle on 4-band multispectral imaging, we provided additional computed images using Zemax software for various incident angles $\theta_{in} = \pm 5^\circ$, $\pm 10^\circ$ and $\pm 15^\circ$. The newly added **Supplementary Figure 13** clearly illustrates that the reflected angle θ_r increases as the incident angle becomes larger. Furthermore, it is evident that at larger incident angles, there is a greater separation between multispectral images in neighboring channels. Notably, when the incident angle increases, the images corresponding to longer wavelengths become increasingly blurred. This blurring effect arises from the strong aberration that occurs at large reflection angles. Conversely, for negative angles of incidence, the images at shorter wavelengths (which are further away from the central wavelength of the meta-mirror) appear blurrier compared to those at longer wavelengths, despite the shorter wavelength channels having smaller angles of reflection. These results indicate that both the intrinsic characteristics of the meta-mirror and optical aberrations play a role in the blurring effect observed in the color images.*

To clarify this point, we revised the text in the main article:

On Page 7 of the main text

*“...Indeed, the blurring effect observed in the color images is influenced by both the intrinsic characteristics of the meta-mirror and optical aberrations, which are also correlated with the incident/reflected angle (refer to **Supplementary Note 2** for more discussions)... ”*

We also added a paragraph and a figure to the Supplementary Material.

Supplementary Note 2: Impact of the incident angle on the 4-band multispectral imaging

*“The incident angle is a critical factor in 4-band multispectral imaging as it affects the reflected angles, lateral separation between color channels, and overall image quality. **Supplementary Figure 13** shows additional computed images for various incident angles, clearly demonstrating that larger incident angles lead to increased reflected angles and greater separation between multispectral images. It is notable that at larger incident angles, the images corresponding to longer wavelengths become more blurred due to the presence of strong aberrations. Conversely, for negative incident angles, the images at shorter wavelengths appear blurrier compared to the longer wavelengths that are closer to the central wavelength of the meta-mirror, despite having smaller angles of reflection. These observations emphasize the collective influence of the incident angle, intrinsic characteristics of the meta-mirror, and optical aberrations on the observed blurring effect in multispectral imaging.”*

Supplementary Figure 13. Calculated images of a number “2” at 4 wavelength bands under various angles of incidence. The θ_{in} and θ_r represent the incident and reflected angles, respectively, for the central wavelength 593 nm.

3. What is designed focal length for the working wavelength of 593 nm? If the focal length varies, what is the impact on the 4-band multispectral imaging?

Reply:

We thank the reviewer for pointing this out. The designed focal length for the wavelength of 593 nm is 7.5 mm. In fact, in the case of a meta-mirror with a fixed footprint, the depth of field (DOF, which is proportional to the square of focal length) and numerical aperture (NA, which is inversely proportional to the focal length) are crucial factors, rather than the focal length itself. Designing the meta-mirror with a larger NA can enhance spatial resolution of the 4-band multispectral imaging. However, a larger NA also results in a smaller DOF. This can pose a challenge when trying to capture 4-band images at the same focal plane. Therefore, when using a focusing meta-mirror with a fixed working area for snapshot multispectral imaging, the focal length should be determined considering the desired DOF and NA.

To clarify this point, we revised the following discussions in the main article.

On Page 6 of the main article

*“...In order to capture a complete set of images in a one-shot measurement, it is necessary to ensure that different color channels corresponding to peak wavelengths in **Fig. 2a** are imaged on a fixed focal plane. Achieving this requires careful consideration of the depth of focus (DOF, which is proportional to the square of focal length) and numerical aperture (NA, which is inversely proportional to the focal length) during the design of the meta-mirror, rather than focusing solely on the focal length. To enable snapshot imaging with acceptable resolution, a focusing meta-mirror with a larger DOF and lower NA is employed. However, this approach involves a trade-off, as it may result in reduced image resolution. In our specific case, we have*

designed a meta-mirror with a NA of 0.02 at a central wavelength of 593 nm to demonstrate snapshot imaging with satisfactory resolution...”

We also modify the caption of Fig. 2 to clarify the designed focal length of the meta-mirror

“...Ray-tracing calculations for the off-axis focusing meta-mirror (designed at a central wavelength of 593 nm with a focal length of 7.5 mm) based on the multi-resonant meta-atoms....”

4. What are the challenges and possible solutions for increasing the number of wavelength bands and the spectral range? Specifically, the authors mentioned that the multi-resonance feature of the developed meta-atom covers from the visible to the near-infrared region, is it possible to increase the number of wavelength bands of the initial multispectral image by designing freeform shaped meta-atoms, such as the method used in the visible light band (Laser & Photonics Reviews, 2022: 210066)?

Reply:

*We thank the reviewer for pointing this out. To achieve multiple resonant peaks in the optical spectrum without the need for interleaving multiple meta-atoms or metasurfaces, it is important to consider the design of the DBR substrate, eigenmodes in the nanostructure, and the coupling between the plasmonic nanostructure and the DBR substrate. The broadest working spectral range of our proposed multispectral imager is naturally determined by the bandwidth of the reflection window of the DBR substrate. This means that the broader the reflection window of the DBR substrate, the wider the range of wavelengths in which the meta-atom could exhibit resonant behavior. In our case, the designed DBR substrate exhibits high reflection intensity toward 930 nm (see the revised **Supplementary Figure 11**), where the longest working wavelength the multi-resonant meta-atom could theoretically achieve resonant behavior.*

To clarify this point, we added the following sentences into the main article

On Page 14

“...While our snapshot hyperspectral system currently demonstrates 18-band data cubes in the visible region, it is worth noting that the number of wavelength bands and the spectral range can be further extended. This is attributed to the multi-resonance feature of the developed meta-atom, which covers a wide range from the visible to the near-infrared region (as shown in **Supplementary Figure 11**). Theoretically, the working spectral range is naturally constrained by the bandwidth of the DBR’s reflection window...”

*We also revised the results shown in **Supplementary Figure 11**.*

Supplementary Figure 11. Optical spectrum of the multi-resonant meta-atom and the DBR substrate. The blue curve shows the numerical LCP-to-RCP reflection spectrum of the designed multi-resonant meta-atom. An Al nano-rod (length = 170 nm, width = 90 nm, thickness = 50 nm, period = 200 nm, thickness of SiO₂ spacer = 135 nm) array standing on a DBR substrate is optimized to possess multiple high-Q resonant peaks across the spectral window from 450 nm to 930 nm. The black curve represents the reflection spectrum of the bare DBR substrate.

In addition, we added the following discussions in the Supplementary Material

Supplementary Note 7: Discussion of the possibility for increasing the number of wavelength bands and the spectral range

“To achieve multiple resonant peaks in the optical spectrum without interleaving multiple meta-atoms or metasurfaces, it is essential to consider the design of the DBR substrate, the eigenmodes present in the nanostructure, and the coupling between the plasmonic nanostructure and the DBR substrate. The spectral range in which the multi-wavelength meta-atom can exhibit resonant behavior is primarily determined by the bandwidth of the reflection window of the DBR substrate. A wider reflection window allows for a broader working spectral range, enabling the meta-atom/MOFM to operate over a greater range of wavelengths...”

*On the other hand, the number of wavelength bands is influenced by two main factors: the number of eigenmodes in the nanostructure and the cavity-like coupling between the nanostructure and DBR substrate. Increasing the number of eigenmodes can be achieved by using freeform nanostructures, as suggested by the referee (Laser & Photonics Reviews 16, 2100663 (2022)). One crucial requirement for freeform nanostructures is that they must possess an anisotropic shape in order to meet the criteria of the geometric phase method for phase modulation. Furthermore, the number of wavelength bands can also be extended by tuning the cavity-like coupling condition. To verify this point, we conducted simulations by increasing the thickness of the dielectric spacer SiO₂ to 2000 nm and 5000 nm (see newly added **Supplementary Figure 16**), while focusing on the same spectral range as demonstrated in the main article. In comparison to the original design with a 135-nm-thick SiO₂ spacer, which generated 4 peak wavelengths from 480 nm to 650 nm, the number of wavelength bands*

increased to 8 and 12 with the 2000-nm-thick and 5000-nm-thick SiO₂ spacers, respectively. Additionally, the multi-resonant peaks were found to be generated in the blue spectral region below 480 nm, covering the missing wavelength bands in the original design.

To address the referee's concern, we added the following discussions in the main article.

On Page 14

“...In fact, both the number of eigenmodes in the nanostructures and the cavity-like coupling between the topmost nanostructures and the DBR substrate play key roles in determining the number of wavelength channels. By utilizing freeform nanostructures with anisotropic shapes^{42, 43}, the number of eigenmodes can be increased, allowing for a greater number of wavelength bands. Moreover, we investigated the impact of tuning the cavity-like coupling condition on the number of wavelength channels by varying the thickness of the SiO₂ spacer, as discussed in **Supplementary Note 7**. In comparison to the original design with a 135-nm-thick SiO₂ spacer, which generated 4 peak wavelengths ranging from 480 nm to 650 nm, the number of wavelength bands increased when SiO₂ spacers increases...”

with two addition references

42. Yang, J., Cui, K., Cai, X., Xiong, J., Zhu, H., Rao, S., *et al.* Ultraspectral imaging based on metasurfaces with freeform shaped meta-atoms. *Laser Photon. Rev.* **16**, 2100663 (2022).
43. Zou, X., Zhang, Y., Lin, R., Gong, G., Wang, S., Zhu, S., *et al.* Pixel-level Bayer-type colour router based on metasurfaces. *Nat. Commun.* **13**, 3288 (2022).

We also added the following discussions in the Supplementary Material

Supplementary Note 7: Discussion of the possibility for increasing the number of wavelength bands and the spectral range

“...The number of wavelength bands in the system is influenced by two key factors: the number of eigenmodes supported by the nanostructure and the cavity-like coupling between the nanostructure and the DBR substrate. Increasing the number of eigenmodes can be achieved by employing freeform nanostructures. Freeform nanostructures^{17, 18}, characterized by their anisotropic shape, adhere to the geometric phase method for phase modulation and enable a higher number of eigenmodes to be supported. Additionally, the number of wavelength bands can be extended by tuning the cavity-like coupling condition. **Supplementary Figure 16a** shows the simulation results of the LCP-to-RCP reflection spectrum for different thicknesses of the dielectric spacer SiO₂. In comparison to the original design with a 135-nm-thick SiO₂ spacer, which produced 4 peak wavelengths ranging from 480 nm to 650 nm (refer to **Supplementary Figure 11**), the number of wavelength bands increased to 8 and 12 with the 2000-nm-thick and 5000-nm-thick SiO₂ spacers, respectively. Notably, the simulations revealed the emergence of multi-resonant peaks in the blue spectral region below 480 nm, effectively filling the previously unexplored wavelength bands in the original design. **Supplementary Figure 16b** presents the simulated phase shift as a function of the structural rotation angle. It is evident from the plot that all the newly generated peak wavelengths align with the expected geometric phase profile. This observation confirms the effectiveness of the proposed method in increasing the number of wavelength channels, which reinforces the potential of the metasurface approach in expanding

the spectral range and enhancing the versatility of multispectral/hyperspectral imaging systems...”

Supplementary Figure 16. Simulated results of the designed meta-atom with 2000-nm-thick and 5000-nm-thick SiO₂ spacers. a Numerical reflection spectrum of the designed meta-atom with different thicknesses of SiO₂. The DBR substrate and physical sizes of the Al nano-rod are the same as those listed in the main article. **b** The circularly cross-polarized phase spectrum as a function of structural rotation angle. All resonant peaks satisfy the geometric phase condition.

with two addition references

17. Yang, J., Cui, K., Cai, X., Xiong, J., Zhu, H., Rao, S., *et al.* Ultraspectral imaging based on metasurfaces with freeform shaped meta-atoms. *Laser Photon. Rev.* **16**, 2100663 (2022).
18. Zou, X., Zhang, Y., Lin, R., Gong, G., Wang, S., Zhu, S., *et al.* Pixel-level Bayer-type colour router based on metasurfaces. *Nat. Commun.* **13**, 3288 (2022).

5. For the training of the Transformer network, the authors only used 40 pairs of images and obtained high-quality results. Why does it only need such a small dataset?

Reply:

We thank the reviewer for pointing this out. As replied in the third comment from Referee 1, we highlight that numerous recent articles have been proposed and successfully demonstrated the training of Transformer with just small data. Some sample articles can be found in the newly added Refs. 38-41 in the main article. In fact, the proposed CODE theory addresses the challenge of small data learning using a completely different philosophy. Simply speaking, typical techniques like discussions in newly added Refs. 38-41 have to force the network to return a good deep solution (as the final solution), while CODE just accepts the weak DE solution. CODE assumes that though the small scale of data results in such a weak solution, the solution itself still contains useful information. Under the assumption, CODE then applies Q-norm to extract the embedded useful information to guide the algorithm as a regularizer, thereby

yielding the final high-quality solution. We refer the referee to Section II.B of Ref. 34 in the revised manuscript for an in-depth discussion about the theoretical aspect of why CODE could work very well even in the absence of big data.

To clarify the above points, we added the following discussions into the main article:
On Page 10

“...It is quite interesting to notice that numerous recent articles have proposed and successfully demonstrated the training of the Transformer with just small data³⁸⁻⁴¹. Remarkably, CODE addresses the challenge of small data learning using a completely different philosophy. Simply speaking, typical techniques³⁸⁻⁴¹ have to force the deep network to return a good deep solution (as the final solution), while CODE just accepts the weak DE solution. CODE assumes that though the small scale of data results in such a weak solution, the solution itself still contains useful information. Under this assumption, CODE then applies Q-norm to extract the embedded useful information to guide the algorithm as a regularizer, thereby yielding the final high-quality solution. We refer interested readers to Ref. [34] for an in-depth discussion about the theoretical aspect of why CODE could work very well even in the absence of big data. Please also see **Supplementary Note 4** for more discussions...”

with four additional references

38. Shao, R., Bi, X. J. Transformers meet small datasets. *IEEE Access* **10**, 118454-118464 (2022).
39. Lee, S. H., Lee, S., Song, B. C. Vision transformer for small-size datasets. 2021. arXiv:2112.13492.
40. Liu, Y., Sangineto, E., Bi, W., Sebe, N., Lepri, B., De Nadai M. Efficient training of visual transformers with small datasets. *Advances in Neural Information Processing Systems* **34**, 23818-23830.
41. Cao, Y.-H., Yu, H., Wu, J. Training vision transformers with only 2040 images. *Computer Vision – ECCV 2022: 17th European Conference, Tel Aviv, Israel, October 23–27, 2022, Proceedings, Part XXV*. Tel Aviv, Israel: Springer-Verlag; 2022. pp. 220–237.

Reviewer: 3

The manuscript presents an experimental demonstration of an 18-channel hyperspectral imaging system. This system utilizes a small form factor of a 4-channel metalens imaging system and employs a small-data convex/deep (CODE) deep learning process. The proposed multi-wavelength off-axis focusing meta-mirror (MOFM) consists of Pancharatnam-Berry (PB) phase Al nanorods and a distributed Bragg reflector (DBR), resulting in multiple channels in reflectance. This characteristic sets it apart from the system described in reference 23. In general, the experimental results align well with the design of the system. The manuscript is well-written and highlights the novel and interesting functionality achieved by incorporating the hot topic of metalens imaging and deep learning in image postprocessing. Overall, with minor revisions addressing the following comments, the manuscript can be accepted for publication.

We thank the reviewer for the positive evaluation of our work and the recommendation for publication after minor revisions.

1. Could the authors provide an explanation for the variation in the circular cross-polarized reflectance spectrum shown in Figure 2b, considering the PB phase scheme where the reflection should be the same under illumination of circularly polarized light?

Reply:

*We thank the reviewer for pointing this out. We agree that in the PB phase scheme, the circular cross-polarized reflection should remain the same regardless of the structural rotation angle. However, it is important to note that when the rotation angle is changed, the distance between neighboring nanostructures is also altered within the fixed period. This change in distance can affect the near-field coupling condition between the nanostructures, potentially leading to slight fluctuations in the reflection intensity. To investigate the influence of the rotation angle on near-field coupling, we provided numerical simulation results and analyzed the electric field distribution at different structural orientation angles ϕ (see revised **Fig. 2b**). As can be seen, the electric field intensity in the gap between neighboring nano-rods becomes slightly stronger at angles of 30° and 45° . This observation confirms that the near-field coupling between neighboring nano-rods becomes stronger at certain rotation angles, resulting in a decrease in the reflection intensity at those angles.*

To address the referee's concern, we added the following sentences into the main article

On Page 6

*“...While the circular cross-polarized reflection in a geometric phase metasurface ideally remains constant irrespective of the structural rotation angle, it is crucial to consider the alteration of the distance between neighboring nanostructures as the rotation angle changes within a fixed period. This change in distance has the potential to influence the near-field coupling condition between the nanostructures (refer to the insets in **Fig. 2b**), resulting in the observed fluctuations in the reflection intensity...”*

We also added the simulated electric field distributions into Fig. 2b in the main article

Figure 2. Design of the multi-resonant meta-atom and off-axis focusing meta-mirror. **a** The numerical reflection spectrum of the designed multi-resonant meta-atom. An Al nano-rod (length = 170 nm, width = 90 nm, thickness = 50 nm, period = 200 nm) array standing on a DBR substrate is optimized to possess four high-Q resonant peaks across the visible window. **b, c** The circular cross-polarized reflection (**b**) and phase (**c**) as a function of structural rotation angle, presenting that the designed multi-resonant meta-atom satisfies the geometric phase conditions at four peak wavelengths. The inset in (**b**) represents the electric field intensity at 593 nm for various structural orientation angles. The inset in (**c**) shows the schematic of the meta-atom. The thickness of dielectric spacer SiO₂ is 135 nm. **d** Ray-tracing calculations for the off-axis focusing meta-mirror (designed at a central wavelength of 593 nm with a focal length of 7.5 mm) based on the multi-resonant meta-atoms. Left images are the spot diagrams. Scale bars: 10 μm. The bottom right images show the phase distributions across the meta-mirror at 593 nm.

- Do the wavelengths of the reflectance resonant peak change when the meta-mirror is illuminated with light at different incidence angles? If yes, there could be a more degree of freedom in terms of wavelength channel in the system.

Reply:

We appreciate the reviewer's comment and suggestion. The coupling between the topmost nano-rod and the bottom DBR substrate enables the generation of multi-wavelength channels. As a result of multiple interference within the constituent layers of the DBR for high reflection, the peak wavelengths can be shifted by changing the incident angle. In order to examine the

sensitivity of the resonant peak wavelengths to the angle of incidence, we conducted numerical simulations and obtained the reflectance spectrum at various incident angles, as shown in **Figure R1**. Consistent with expectations, the resonant peak wavelengths exhibit slight blue-shifts with increasing incident angle. To utilize the incident angle as a parameter for wavelength channel design, the peak wavelength shift should exceed the full width at half maximum (FWHM) of the peaks under normal illumination. This condition can be achieved when the incident angle is greater than 15° . However, it is worth noting that at an incident angle of 15° , the reflection intensity of individual peaks decreases, particularly for the fourth wavelength channel, which drops to approximately 60%. Therefore, modulating the resonant wavelengths by varying the incidence angle is deemed unsuitable in this case.

Figure R1. LCP-to-RCP reflection spectrum at various angles of incidence. The incident angle was varied from 0° to 15° . To provide a clear representation of the peak wavelength shift, a zoomed-in image of the spectral window ranging from 505 nm to 545 nm is presented in the right panel.

- In line 133, the authors mention that the geometric shape of the meta-atom is randomly chosen. However, considering the PB phase scheme, it is expected that the meta-atom should be optimized to achieve maximum circular cross-polarized reflectance at the operation wavelength. Could the authors provide a comment on this apparent contradiction?

Reply:

We thank the reviewer for this comment. We apologize for any confusion caused. The main point we would like to emphasize is that arbitrary nanostructures with anisotropic shapes have the potential to be utilized for the PB phase, as long as their geometric dimensions are optimized to achieve the maximum circular cross-polarized reflection within the desired spectral range. To clarify this statement, we revised the following sentences in the main article.

“...Indeed, the selection of the geometric shape for the meta-atom in our study was random and primarily intended for illustrative purposes. The key concept is that other anisotropic nanostructures can be employed to attain similar results and outcomes, as long as their physical dimensions are carefully optimized to maximize the efficiency of circular polarization conversion...”

4. Could the authors provide an explanation for the discrepancy in color between Figure 3c and Figure 3d, where the image at a wavelength of 593 nm (channel 3) appears yellow in Figure 3c but red in Figure 3d?

Reply:

*We thank the reviewer for bringing up this point. The discrepancy in color between the measured images shown in **Figs. 3c** and **3d** can be attributed to the white balance using different light sources. For simplicity, a white LED was used for white balance during the capture of the number "2" images (**Fig. 3c**), while a projector with a white background was used for white balance before capturing the color potted flower image (**Fig. 3d**). As shown in the newly added **Supplementary Figure 14**, the spectral distribution of light intensity differs between these two light sources. Specifically, the intensity of the projector in the green color range is relatively lower compared to the blue and red colors. As a result, the image at channel 3 (593 nm) in **Fig. 3d** appears reddish. Conversely, the white LED demonstrates more consistent intensity in the wavelength range from 510 nm to 650 nm, corresponding to the green to red color range, making the color at channel 3 in **Fig. 3c** closer to its actual color. However, it is important to emphasize that the observed color discrepancy does not impact the accuracy of the hyperspectral imaging results. This is because the wavelength channels of the multispectral images are spatially distributed in free space. Each color image captured at the correct spatial position corresponds to the respective wavelength channel. Furthermore, the hyperspectral imaging data cubes are obtained using the developed small-data learning theory, which reconstructs individual color images at each wavelength band based on the CIE 1931 color space. Therefore, the accuracy of the hyperspectral imaging results remains unaffected by the observed color differences between the images.*

To clarify this point, we added the following discussions in the main article.

On Page 8

“...One can see the discrepancy in color at channel 3 between the measured images shown in **Figs. 3c** and **3d**, which can be attributed to the use of different light sources for white balance. However, we emphasize that this variation in color does not impact the accuracy of the multispectral/hyperspectral imaging results. This is because the spatial distribution of wavelength channels ensures that each color image corresponds to its respective wavelength channel at the correct spatial position (refer to **Supplementary Note 3** for more discussions).”

We also added the following discussions and a figure into the Supplementary Material.

Supplementary Note 3: Light sources used for white balance

“**Supplementary Figure 14** provides a comparison of the optical spectra emitted by two different light sources used for white balance. The projector shows a lower intensity in the green color range compared to the blue and red colors, causing channel 3 (593 nm) in **Fig. 3d** to appear more reddish. On the other hand, the white LED exhibits a more consistent intensity across the wavelength range corresponding to green to red colors, resulting in the color at channel 3 in **Fig. 3c** being closer to its true representation. Importantly, we emphasize that the observed color discrepancy does not compromise the accuracy of the multispectral/hyperspectral imaging results. This is because the spatial distribution of wavelength channels in free space ensures that each captured color image corresponds to the specific wavelength channel at its designated spatial position. Additionally, the small-data learning theory employed for obtaining the hyperspectral imaging data cubes reconstructs individual color images for each wavelength band based on the CIE 1931 color space, guaranteeing accuracy in the results regardless of any color differences observed between the images.”

Supplementary Figure 14. The spectrum of different light sources. The intensity spectrum of the projector is represented by the blue curve, while the white LED is represented by the olive curve. A white background is used when measuring the optical spectrum of the projector.

- In lines 184-186, the authors state that the images shown in Figure 3d are blurrier compared to those presented in Figure 3c due to the low Q-factor of the MOFM. Could the authors provide information on the bandwidth of the light beam from the acousto-optic tunable filter (AOTF)?

Reply:

*We thank the reviewer for this suggestion. In the newly added **Supplementary Figure 2c**, we presented the light intensity spectrum of the laser beams generated by the AOTF. Additionally, in **Supplementary Figure 2d**, we provided the peak-fitting FWHM of the light beams. These measurements allow us to assess the bandwidth of the laser beams and compare it with the bandwidth of the MOFM, as shown in **Fig. 3a** and **Supplementary Figures 2c** and **2d**. The*

results demonstrate that the bandwidth of the laser beams from the AOTF is significantly smaller than the bandwidth of the MOFM, thereby confirming the assertion made in the main article.

To verify this point, we added the following sentences in the main article.

On Page 8

“...we can see that the images at the four channels shown in **Fig. 3d** are blurrier than those presented in **Fig. 3c**, which resulted from the relatively low Q-factor of the measured peaks (see **Fig. 3a**, **Supplementary Figures 2c** and **2d** for comparison)...”

We also added the bandwidth analysis for the light beam generated by the AOTF into the **Supplementary Figure 2**

Supplementary Figure 2. Optical setup for the imaging characterization of multi-wavelength off-axis focusing meta-mirror. a A supercontinuum laser (NKT Photonics FIU-15) combined with an acousto-optic tunable filter (AOTF, SuperK SELECT) is utilized to select the

wavelength in the visible. In this case, resolution targets with different-shaped apertures are used as the object. **b** To demonstrate the snapshot hyperspectral imaging capability of the MOFM, full-color images from a projector are used as the objects. A half-wave plate (Thorlabs AHWP05M-600), a linear polarizer (Thorlabs LPVISE100-A), and a quarter-wave plate (Thorlabs AQWP05M-600) are used to determine the polarization state. M: mirror; I: iris; $\lambda/2$: half-wave plate; P: linear polarizer; $\lambda/4$: quarter-wave plate; O: objective (Mitutoyo 10 \times magnification with 0.28 numerical aperture); RT: resolution targets; L: Lens. **c** The optical spectrum of the laser beams emitted by the AOTF. The wavelength range shown is limited to 530-650 nm due to the capabilities of the used micro-spectrometer (Phekda Series, PD) for measurement. **d** The peak-fitting FWHM of the laser beams carried out from (c).

6. The authors may consider addressing certain limitations in their manuscript. For instance, they can discuss the wavelength resolution of the ground truth, which is mentioned to be 10 nm. In Figure 5a, it is observed that the petal appears in channels ranging from 610 nm to 650 nm, likely due to the broadband nature of the red pixel in the projector. However, what if the red color of the petal is narrowband in nature, such as 10 nm or 20 nm? In such cases, can the CODE generate appropriate 18 channels of hyperspectral images if the petal (or any other color image feature) is only present in a single ground truth channel? It would be valuable for the authors to discuss the limitations of the spectral resolution in their system and explore possibilities for improvement in the discussion section.

Reply:

*We thank the reviewer for this comment. Based on the developed CODE theory, to maintain image quality, we recommend setting the number of bands in the output hyperspectral image to 4.5 times the number of bands in the input image, as demonstrated in this study. In order to improve wavelength resolution, we propose modifying the design of multi-resonant meta-atoms to increase the number of bands in the multispectral image. This can be achieved by adjusting the cavity-like coupling effect between the topmost nano-rod and the underlying DBR substrate. As can be seen in the newly added **Supplementary Figure 16a**, increase of the SiO₂ thickness between the nano-rod and DBR from 135 nm to 2000 nm results in an increase in the number of resonant wavelengths from 4 to 8 within the wavelength range of 480 nm to 650 nm. In addition, further increasing the SiO₂ thickness to 5000 nm can generate 12 resonant wavelengths within the same wavelength range. Furthermore, **Supplementary Figure 16b** provides verification that all resonant peak wavelengths satisfy the PB phase condition. These findings further support the conclusion that adjusting the thickness of the dielectric spacer between the metal structure and DBR effectively enhances the meta-mirror design for multispectral imaging, resulting in an increased number of wavelength channels. Therefore, by utilizing the CODE theory, it becomes feasible to generate hyperspectral images with 54 wavelength bands, leading to an expected improvement in wavelength resolution from approximately 10 nm to 3.9 nm. However, achieving this outcome necessitates providing the training model with ground truth data that corresponds to the specific number of bands. In other words, when converting a multispectral image with 12 bands into a hyperspectral image with 54 bands using CODE theory, it is essential to supply 54-band ground truth images for model training. This practical implementation aspect presents challenges, primarily due to the requirement of obtaining accurate ground truth data using multiple bandpass filters. Therefore, while striving to enhance spectral resolution, it is crucial to*

consider the difficulties associated with acquiring precise ground truth data in order to maintain a balanced approach.

On the other hand, we would like to point out that the CODE theory is capable of extracting the spectral information of pixel *A*, which has a bandwidth of approximately 10 nm, despite each wavelength channel in the multispectral image containing data with a bandwidth of around 20-30 nm (as shown in **Fig. 3a** in the main article). This capability arises from the CODE theory learning the mapping between the known hyperspectral ground truth and its corresponding multispectral input, which is the multispectral image generated by the MOFM. By utilizing this learned mapping, the CODE theory can effectively infer the unknown hyperspectral images that correspond to the captured multispectral image. Therefore, even if pixel *A* is only present in a subset or a single spectral band within the ground truth data library, its hyperspectral information can still be accurately inferred through the learned mapping from other data pairs. This aspect is further supported by our research findings. As demonstrated in **Figs. 3d** and **5** in the main article, the original multispectral image lacked meaningful information in the blue region (wavelengths below 500 nm). However, through the application of the CODE theory to construct a hyperspectral image, we were able to obtain highly accurate blue light information that closely aligned with the ground truth. This serves as evidence that certain pixels within the captured object, whose spectral information may be concealed within a wide spectral band image or cannot be effectively extracted by the MOFM, can still be precisely recovered utilizing the small-data learning theory developed in this study.

To discuss the limitation and improvement of the spectral resolution in our system, we added the following discussions into the main article.

On Page 13

“...As shown in **Supplementary Figure 16**, increasing the dielectric spacer thickness to 5000 nm allows for the generation of 12 resonant wavelengths within the same spectral range, which reveals the possibility to further enhance the spectral resolution. To preserve the fidelity of the image throughout the conversion process, we propose adhering to a ratio of 4.5 for the number of bands in the output hyperspectral image compared to the number of bands in the input multispectral image, based on the principles of the CODE theory showcased in this study. Thus, a 12-band multispectral imaging theoretically enables the generation of hyperspectral images with 54 wavelength bands, resulting in an improved wavelength resolution from approximately 10 nm to 3.9 nm. However, it is important to note that this requires providing the training model with ground truth data that corresponds to the specific number of bands. Practical implementation of this approach poses challenges, particularly in obtaining accurate ground truth data through multiple bandpass filters. Therefore, it is crucial to consider the difficulties associated with acquiring precise ground truth data while striving to enhance spectral resolution, and maintain a balanced approach throughout the process. These findings highlight the potential for further extending the number of wavelength bands and spectral range by optimizing the design parameters and cavity-like coupling conditions in future iterations of the system.

Furthermore, we highlight that the CODE theory exhibits the capability to extract the spectral information of an arbitrary pixel *A*, which possesses a bandwidth of approximately 10 nm, despite the fact that each wavelength channel in the multispectral image contains data with a broader bandwidth of around 20-30 nm (as shown in **Fig. 3a**). This ability stems from the CODE theory's acquisition of the mapping between the known hyperspectral ground truth and its corresponding input, which is the multispectral image generated by the MOFM. By leveraging

this learned mapping, the CODE theory can effectively infer the unknown hyperspectral images that correspond to the multispectral image. Consequently, even if pixel A is exclusively present in a subset or a single spectral band within the ground truth data library, its hyperspectral information can still be accurately deduced through the utilization of the learned mapping from other data pairs....”

We also added the following discussions in the Supplementary Material

Supplementary Note 7: Discussion of the possibility for increasing the number of wavelength bands and the spectral range

“...The number of wavelength bands in the system is influenced by two key factors: the number of eigenmodes supported by the nanostructure and the cavity-like coupling between the nanostructure and the DBR substrate. Increasing the number of eigenmodes can be achieved by employing freeform nanostructures. Freeform nanostructures^{17, 18}, characterized by their anisotropic shape, adhere to the geometric phase method for phase modulation and enable a higher number of eigenmodes to be supported. Additionally, the number of wavelength bands can be extended by tuning the cavity-like coupling condition. **Supplementary Figure 16a** shows the simulation results of the LCP-to-RCP reflection spectrum for different thicknesses of the dielectric spacer SiO₂. In comparison to the original design with a 135-nm-thick SiO₂ spacer, which produced 4 peak wavelengths ranging from 480 nm to 650 nm (refer to **Supplementary Figure 11**), the number of wavelength bands increased to 8 and 12 with the 2000-nm-thick and 5000-nm-thick SiO₂ spacers, respectively. Notably, the simulations revealed the emergence of multi-resonant peaks in the blue spectral region below 480 nm, effectively filling the previously unexplored wavelength bands in the original design. **Supplementary Figure 16b** presents the simulated phase shift as a function of the structural rotation angle. It is evident from the plot that all the newly generated peak wavelengths align with the expected geometric phase profile. This observation confirms the effectiveness of the proposed method in increasing the number of wavelength channels, which reinforces the potential of the metasurface approach in expanding the spectral range and enhancing the versatility of multispectral/hyperspectral imaging systems...”

Supplementary Figure 16. Simulated results of the designed meta-atom with 2000-nm-thick and 5000-nm-thick SiO₂ spacers. a Numerical reflection spectrum of the designed meta-atom with different thicknesses of SiO₂. The DBR substrate and physical sizes of the Al nano-rod are

the same as those listed in the main article. **b** The circularly cross-polarized phase spectrum as a function of structural rotation angle. All resonant peaks satisfy the geometric phase condition.

with two addition references

17. Yang, J., Cui, K., Cai, X., Xiong, J., Zhu, H., Rao, S., *et al.* Ultraspectral imaging based on metasurfaces with freeform shaped meta-atoms. *Laser Photon. Rev.* **16**, 2100663 (2022).
18. Zou, X., Zhang, Y., Lin, R., Gong, G., Wang, S., Zhu, S., *et al.* Pixel-level Bayer-type colour router based on metasurfaces. *Nat. Commun.* **13**, 3288 (2022).

Reviewer #2 (Remarks to the Author):

I thank the authors for the final corrections and congratulate them on their interesting work.

Reviewer #3 (Remarks to the Author):

Drawing upon my expertise in metasurfaces, I would like to note that the authors have adeptly tackled the technical issues and concerns highlighted by Reviewer 2 and myself in our prior review. The experimental results harmoniously correlate with the theoretical computations and the underlying design. The manuscript is well-written, accentuating the inventive, captivating, and valuable capabilities attained through the integration of the cutting-edge subjects of metalens imaging and deep learning within image postprocessing. In summation, I enthusiastically endorse the publication of this manuscript in Nature Communications.